# WHY HIGH-RANK NEURAL NETWORKS GENERALIZE?: AN ALGEBRAIC FRAMEWORK WITH RKHSS

**Yuka Hashimoto**[1,2], **Sho Sonoda**[2,3], **Isao Ishikawa**[4,2], **Masahiro Ikeda**[5,2]
1. NTT, Inc., 2. RIKEN AIP, 3. CyberAgent, Inc., 4. Kyto University, 5. The University of Osaka

## ABSTRACT

We derive a new Rademacher complexity bound for deep neural networks using Koopman operators, group representations, and reproducing kernel Hilbert spaces (RKHSs). The proposed bound describes why the models with high-rank weight matrices generalize well. Although there are existing bounds that attempt to describe this phenomenon, these existing bounds can be applied to limited types of models. We introduce an algebraic representation of neural networks and a kernel function to construct an RKHS to derive a bound for a wider range of realistic models. This work paves the way for the Koopman-based theory for Rademacher complexity bounds to be valid for more practical situations.

## 1 INTRODUCTION

Understanding the generalization property of deep neural networks has been one of the biggest challenges in the machine learning community. The generalization property describes how the model can fit unseen data. Classically, the generalization error is bounded using the VC-dimension theory (Harvey et al., 2017; Anthony & Bartlett, 2009). Norm-based (Neyshabur et al., 2015; Bartlett et al., 2017; Golowich et al., 2018; Neyshabur et al., 2018; Wei & Ma, 2019; Li et al., 2021; Ju et al., 2022; Weinan E et al., 2022) and compression-based (Arora et al., 2018; Suzuki et al., 2020) bounds have also been investigated. The norm-based bounds depend on the matrix $(p, q)$ norm of the weight matrices, and the compression-based bounds are derived by investigating how much the networks can be compressed. These bounds imply that low-rank weight matrices and weight matrices with small singular values, i.e., nearly low-rank matrices, have good effects for generalization. See Appendix C for more details about the existing bounds.

On the other hand, phenomena in which models with weight matrices that are high-rank and have large singular values generalize well have been empirically observed (Goldblum et al., 2020). Since the norm-based and compression-based bounds focus only on the low-rank and nearly low-rank cases, they cannot describe these phenomena. To theoretically describe these phenomena, the Koopman-based bound was proposed (Hashimoto et al., 2024). Koopman operators are linear operators that describe the compositions of functions, which are essential structures of neural networks. This existing bound is described by the ratio of the norm to the determinant of each weight matrix as

$$O\left(\prod_{l=1}^{L} \frac{G_l \|K_{\sigma_l}\|_{H_l} \|W_l\|^{s_{l-1}}}{\sqrt{S} \det(W_l^* W_l)^{1/4}}\right), \tag{1}$$

where $S$ is the sample size, $s_l$ represents the smoothness of the $l$th layer, $G_l$ is a factor determined by the $l \sim L$th layers, $K_{\sigma_l}$ is the Koopman operator with respect to the activation function $\sigma_l$, and $\|\cdot\|_{H_l}$ represents the operator norm in a Sobolev space $H_l$. Since the determinant factor appears in the denominator of the bound, even if the weight matrices are high rank and have large singular values, this bound can be small. The Koopman-based bound theoretically sheds light on why neural networks with high-rank weight matrices generalize well.

However, the existing analysis for the Koopman-based bound strongly depends on the smoothness of models and the unboundedness of the data space, which excludes realistic models with bounded data space and with activation functions such as the hyperbolic tangent, sigmoid, and ReLU-type nonsmooth functions. In addition, the dependency of the bound on the activation function is not clear. In fact, the factors $\|K_{\sigma_l}\|_{H_l}$ and $G_l$ in the bound (1) is hard to evaluate in many cases.

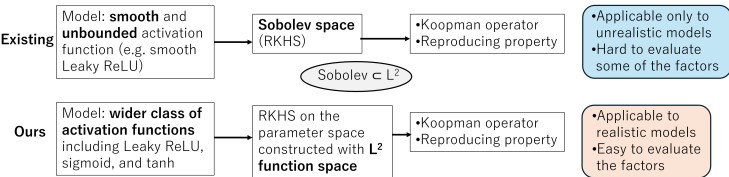

Figure 1: Summary of the framework of the existing and proposed Koopman-based bounds

In this paper, we propose a new Koopman-based bound that resolves the issues of the existing Koopman-based bounds. The proposed bound is described as

$$O\bigg(\prod_{l=1}^{L} \frac{G_l \|K_{\sigma_l}\|_{\mathcal{L}_l}}{\sqrt{S}\det(W_l^* W_l)^{1/4}}\bigg),$$

where $\|\cdot\|_{\mathcal{L}_l}$ is the operator norm in a $L^2$ function space. Similar to the existing Koopman-based bounds, the proposed bound describes why high-rank neural networks generalize well. On the other hand, the difference of the function space $\mathcal{L}_l$ from $H_l$ gives a significant benefit to the proposed bound. We note that $\mathcal{L}_l$ is larger than $H_l$, and $\mathcal{L}_l$ enables us to analyze nonsmooth deep models and bounded data space. In addition, it enables us to evaluate the factors $\|K_{\sigma_l}\|_{\mathcal{L}_l}$ and $G_l$ easily (see Lemmas 2.3–2.5) and understand the effect of the activation functions on the deep model. As a result, the proposed bound significantly improves the existing bound in the sense that it can be applied to a wider range of models and enables us to understand the models well.

To achieve the above improvement, we introduce a kernel function defined on the parameter space using linear operators on a Hilbert space to which models belong. This kernel function allows us to construct a reproducing kernel Hilbert space (RKHS) that describes realistic deep models with nonsmooth activation function and bounded data space. We use the Rademacher complexity to derive generalization bounds. The Rademacher complexity measures the complexity of the model, which also describe the generalization property. Using the reproducing property of the RKHS, we can bound the Rademacher complexity with the operator norms of the linear operators. For linear operators, we use group representations and Koopman operators. We first focus on algebraic representations of models using group representations. A typical example is the representation of the affine group, which describes invertible neural networks. We then focus on representations using Koopman operators with respect to the weight matrices, which describe neural networks with non-constant width. We schematically show the summary of the framework of the existing and proposed Koopman-based bounds in Figure 1.

The main contributions of this paper are as follows:
- We introduce an algebraic representation of models that can represent deep neural networks as typical examples. To describe the action of parameters on models, we focus on group representations, which enables us to represent invertible neural networks, and Koopman operators, which enables us to represent more general neural networks (Subsections 3.1 and 5.1).
- We define a kernel function to construct an RKHS that describes the model. We derive a new Rademacher complexity bound using this kernel (Subsection 3.2). The proposed bound describes why the models with high-rank weight matrices generalize well for a wider range of models than the existing bounds (Section 4 and Subsections 5.2–5.4).

**Notations and remarks**    For $d \in \mathbb{N}$ and a Lebesgue measure space $\mathcal{X} \subseteq \mathbb{R}^d$, let $L^2(\mathcal{X})$ be the space of complex-valued squared Lebesgue-integrable functions on $\mathcal{X}$. We denote by $\mu_{\mathcal{X}}$ the Lebesgue measure on $\mathcal{X}$. For a Hilbert space $\mathcal{H}$, let $\langle\cdot,\cdot\rangle_{\mathcal{H}}$ be the inner product in $\mathcal{H}$. We omit the subscript $\mathcal{H}$ when it is obvious. We denote by $B(\mathcal{H}_1, \mathcal{H}_2)$ be the space of bounded linear operators from $\mathcal{H}_1$ to $\mathcal{H}_2$. In particular, we denote $B(\mathcal{H}, \mathcal{H}) = B(\mathcal{H})$. All the technical proofs are in Appendix A.

## 2    PRELIMINARIES

### 2.1    KOOPMAN OPERATOR

Koopman operator is a linear operator that represents the composition of nonlinear functions. Since neural networks are constructed using compositions, Koopman operators play an essential role in

analyzing neural networks. Let $\mathcal{X} \subseteq \mathbb{R}^d$ be a Lebesgue measure space. Koopman operators are defined as follows. We also introduce weighted Koopman operator, which is a generalization of Koopman operator.

**Definition 2.1** (Koopman operator and weighted Koopman operator). Let $\tilde{\mathcal{X}} \subseteq \mathbb{R}^{d_1}$ and $\mathcal{X} \subseteq \mathbb{R}^{d_2}$. The *Koopman operator* $K_\sigma$ with respect to a map $\sigma : \tilde{\mathcal{X}} \to \mathcal{X}$ is a linear operator from $L^2(\mathcal{X})$ to $L^2(\tilde{\mathcal{X}})$ that is defined as $K_\sigma h(x) = h(\sigma(x))$ for $h \in L^2(\tilde{\mathcal{X}})$. In addition, the *weighted Koopman operator* $\tilde{K}_{\psi,\sigma}$ with respect to maps $\psi : \tilde{\mathcal{X}} \to \mathbb{C}$ and $\sigma : \tilde{\mathcal{X}} \to \mathcal{X}$ is a linear operator from $L^2(\tilde{\mathcal{X}})$ to $L^2(\mathcal{X})$ that is defined as $\tilde{K}_{\psi,\sigma} h(x) = \psi(x)h(\sigma(x))$ for $h \in L^2(\mathcal{X})$.

We will consider the Koopman operators with respect to activation functions. Throughout this paper, we assume these Koopman operators are bounded.

**Assumption 2.2** (Boundedness of Koopman operators). *The Koopman operator $K_\sigma$ with respect to a map $\sigma$ is bounded, i.e., the operator norm defined as $\|K_\sigma\| = \sup_{\|h\|=1} \|K_\sigma h\|$ is finite.*

Indeed, we have the following lemma regarding the sufficient condition of the boundedness of Koopman operators.

**Lemma 2.3.** *Assume $\sigma : \tilde{\mathcal{X}} \to \mathcal{X}$ is bijective, $\sigma^{-1}$ is differentiable, and the Jacobian of $\sigma^{-1}$ is bounded in $\mathcal{X}$. Then, we have $\|K_\sigma\| \leq \sup_{x \in \mathcal{X}} |J\sigma^{-1}(x)|^{1/2}$, where $J\sigma^{-1}$ is the Jacobian of $\sigma^{-1}$. In particular, the Koopman operator $K_\sigma$ is bounded.*

The following lemma is regarding the boundedness of well-known elementwise activation functions defined as $\sigma([x_1, \ldots, x_d]) = [\tilde{\sigma}(x_1), \cdots, \tilde{\sigma}(x_d)]$ for a map $\tilde{\sigma} : \mathbb{R} \to \mathbb{R}$.

**Lemma 2.4.** *Let $\tilde{\mathcal{X}} = [a_1, b_1] \times \cdots \times [a_d, b_d] \subseteq \mathbb{R}^d$ be a bounded rectanglar domain, and let $\mathcal{X} = \sigma(\tilde{\mathcal{X}})$. If $\sigma$ is the elementwise hyperbolic tangent defined as $\tilde{\sigma}(x) = \tanh(x)$, then we have $\mathcal{X} \subset [-1, 1]^d$ and $\|K_\sigma\| \leq (\prod_{i=1}^d \sup_{x \in \tilde{\sigma}([a_i, b_i])} 1/(1 - x^2))^{1/2}$. If $\sigma$ is the elementwise sigmoid defined as $\tilde{\sigma}(x) = 1/(1 + e^{-x})$, then we have $\mathcal{X} \subset [-1, 1]^d$ and $\|K_\sigma\| \leq (\prod_{i=1}^d \sup_{x \in \tilde{\sigma}([a_i, b_i])} 1/(x - x^2))^{1/2}$.*

Even if $\sigma$ is not differentiable, the Koopman operator is bounded, and we can evaluate the upper bound in some cases.

**Lemma 2.5.** *Let $\tilde{\mathcal{X}} = \mathcal{X} = \mathbb{R}^d$. Let $\sigma$ be the elementwise Leaky ReLU defined as $\tilde{\sigma}(x) = ax$ for $x \leq 0$ and $\tilde{\sigma}(x) = x$ for $x > 0$, where $a > 0$. Then, we have $\|K_\sigma\| \leq \max\{1, 1/a^d\}^{1/2}$.*

## 2.2 REPRODUCING KERNEL HILBERT SPACE (RKHS)

In addition to the $L^2$ function space, we also consider reproducing kernel Hilbert spaces. Let $\Theta$ be a non-empty set for parameters. We first introduce positive definite kernel.

**Definition 2.6** (Positive definite kernel). A map $k : \Theta \times \Theta \to \mathbb{C}$ is called a *positive definite kernel* if it satisfies the following conditions:

- $k(\theta_1, \theta_2) = \overline{k(\theta_2, \theta_1)}$ for $\theta_1, \theta_2 \in \Theta$,
- $\sum_{i,j=1}^n \overline{c_i} c_j k(\theta_i, \theta_j) \geq 0$ for $n \in \mathbb{N}$, $c_i \in \mathbb{C}$, and $\theta_i \in \Theta$.

Let $\phi : \Theta \to \mathbb{C}^\Theta$ be the *feature map* associated with $k$, defined as $\phi(\theta) = k(\cdot, \theta)$ for $\theta \in \Theta$ and let $\mathcal{R}_{k,0} = \{\sum_{i=1}^n \phi(\theta_i)c_i \mid n \in \mathbb{N}, c_i \in \mathbb{C}, \theta_i \in \Theta \ (i = 1, \ldots, n)\}$. We can define a map $\langle \cdot, \cdot \rangle_{\mathcal{R}_k} : \mathcal{R}_{k,0} \times \mathcal{R}_{k,0} \to \mathbb{C}$ as

$$\left\langle \sum_{i=1}^n \phi(\theta_i)c_i, \sum_{j=1}^m \phi(\xi_j)d_j \right\rangle_{\mathcal{R}_k} = \sum_{i=1}^n \sum_{j=1}^m \overline{c_i} d_j k(\theta_i, \xi_j).$$

The *reproducing kernel Hilbert space (RKHS)* $\mathcal{R}_k$ associated with $k$ is defined as the completion of $\mathcal{R}_{k,0}$. One important property of RKHSs is the reproducing property $\langle \phi(\theta), f \rangle_{\mathcal{R}_k} = f(\theta)$ for $f \in \mathcal{R}_k$ and $\theta \in \Theta$, which is also useful for deriving a Rademacher complexity bound.

### 2.3 Group representation

Group representation is also a useful tool to analyze the deep structure of neural networks (Sonoda et al., 2025). Let $G$ be a locally compact group. A *unitary representation* $\rho : G \to B(\mathcal{H})$ for a Hilbert space $\mathcal{H}$ is a map whose image is in the space of unitary operators on $\mathcal{H}$, that satisfies $\rho(g_1 g_2) = \rho(g_1)\rho(g_2)$ and $\rho(g_1^{-1}) = \rho(g_1)^*$ for $g_1, g_2 \in G$, and for which $g \mapsto \rho(g)h$ is continuous for any $h \in \mathcal{H}$. Here, $^*$ means the adjoint. If there exists no nontrivial subspace $\mathcal{M}$ of $\mathcal{H}$ such that $\rho(g)\mathcal{M} \subseteq \mathcal{M}$ for any $g \in G$, then the representation $\rho$ is called *irreducible*.

For irreducible unitary representations, we have the following fundamental result (see, e.g. Folland (1995, Lemma 3.5)), which we will apply to show the universality of the model. Here, the commutant of a subset $\mathcal{A} \subseteq B(\mathcal{H})$ is defined as the set $\{A \in B(\mathcal{H}) \mid AB = BA \text{ for } B \in \mathcal{A}\}$.

**Lemma 2.7** (Schur's lemma). *A unitary representation $\rho$ of $G$ is irreducible if and only if the commutant of $\rho(G)$ contains only scalar multiples of the identity.*

We also apply the following fundamental result (see, e.g., Davidson (1996, Theorem I.7.1)).

**Lemma 2.8** (von Neumann double commutant theorem). *Let $\mathcal{A}$ be a subalgebra of $B(\mathcal{H})$ that satisfies "$A \in \mathcal{A} \Rightarrow A^* \in \mathcal{A}$" and is closed with respect to the operator norm. Then, the double commutant (i.e., the commutant of the commutant) of $\mathcal{A}$ is equal to the closure of $\mathcal{A}$ with respect to the strong operator topology.*

## 3 Problem Setting

We formulate deep models, which include the neural network model as a special example, using operators. Then, we define an RKHS to analyze the deep model.

### 3.1 Algebraic representation of deep models with group representations

Let $G$ be a locally compact group and $\rho : G \to B(\mathcal{H})$ be a unitary representation on a Hilbert space $\mathcal{H}$. We consider an algebraic representation of $L$-layered deep model in $\mathcal{H}$

$$f(g_1, \ldots, g_L) = \rho(g_1)A_1\rho(g_2)A_2 \cdots A_{L-1}\rho(g_L)v, \tag{2}$$

where $g_1, \ldots, g_L \in G$ are learnable parameters, $A_1, \ldots, A_L \in B(\mathcal{H})$ and $v \in \mathcal{H}$ are fixed.

**Example 3.1** (Scaled neural network with invertible weights). Let $G = GL(d) \ltimes \mathbb{R}^d$ be the affine group and $\mathcal{H} = L^2(\mathbb{R}^d)$. Here, $GL(d)$ is the group of $d$ by $d$ invertible matrices. Let $\rho : G \to B(\mathcal{H})$ be the representation of $G$ on $\mathcal{H}$ defined as $\rho(g)h(x) = |\det W|^{1/2}h(W(x-b))$ for $g = (W, b) \in G$, $h \in L^2(\mathbb{R}^d)$, and $x \in \mathbb{R}^d$. Note that $\rho$ is an irreducible unitary representation. In addition, let $v \in L^2(\mathbb{R}^d)$ be the final nonlinear transformation, $\sigma_l : \mathbb{R}^d \to \mathbb{R}^d$ be an activation function satisfying Assumption 2.2, and $A_l = K_{\sigma_l}$ be the Koopman operator with respect to $\sigma_l$ for $l = 1, \ldots, L-1$. For example, $\sigma_l$ is the elementwise Leaky ReLU. Then, the deep model (2) is

$$f(g_1, \ldots, g_L)(x) = v(W_L \sigma_{L-1}(W_{L-1} \cdots \sigma_1(W_1 x - W_1 b_1) \cdots - W_{L-1}b_{L-1}) - W_L b_L)$$
$$\times |\det W_1|^{1/2} \cdots |\det W_{L-1}|^{1/2}.$$

**Example 3.2** (Deep model with new structures). In addition to describing existing neural networks, we can develop a new model using the abstract model (2). Let $G = \{(a, b, c) \mid a, b \in \mathbb{R}^d, \ c \in \mathbb{R}\}$ be the Heisenberg group (Thangavelu, 1998). The product in $G$ is defined as $(a_1, b_1, c_1) \cdot (a_2, b_2, c_2) = (a_1 + a_2, b_1 + b_2, 1/2 \langle a_1, b_2 \rangle)$, where $\langle a_1, b_2 \rangle$ is the Euclidean inner product of $a_1$ and $a_2$. Let $\mathcal{H} = L^2(\mathbb{R}^d)$ and $\rho : G \to B(\mathcal{H})$ be the representation of $G$ on $\mathcal{H}$ defined as $\rho(g)h(x) = e^{i(c-1/2\langle a,b \rangle)}e^{i\langle a,x \rangle}h(x - b)$ for $g = (a, b, c)$, where i is the imaginary unit. Note that $\rho$ is an irreducible unitary representation. Let $v$ and $A_l$ be the same as in Example 3.1. Then, the deep model (2) is

$$f(g_1, \ldots, g_L)(x) = e^{i(c_1 - \langle a_1, b_1 \rangle/2)} \cdots e^{i(c_L - \langle a_L, b_L \rangle/2)}$$
$$\cdot e^{i\langle a_1, x \rangle}e^{i\langle a_2, \sigma_1(x-b_1) \rangle} \cdots e^{i\langle a_L, \sigma_{L-1}(\sigma_{L-2}(\cdots \sigma_1(x-b_1)\cdots -b_{L-2})-b_{L-1}) \rangle}$$
$$\cdot v(\sigma_{L-1}(\cdots \sigma_1(x - b_1) - b_{L-1}) - b_L).$$

Instead of directly considering the model (2), we focus on the following regularized model with a parameter $c > 0$ on a data space $\mathcal{X}_0$:

$$F_c(g_1, \ldots, g_L, x) = \langle \rho(g_1)A_1\rho(g_2)A_2 \cdots A_{L-1}\rho(g_L)v, p_{c,x} \rangle, \tag{3}$$

where $p_{c,x} \in \mathcal{H}$ for $c > 0$ and $x \in \mathcal{X}_0$. We assume for any $c > 0$, there exists $E(c) > 0$ such that $\|p_{c,x}\|^2 \leq E(c)$. This regularization is required to technically derive the Rademacher complexity bound using the framework of RKHSs. However, as the following example indicates, the regularized model (3) sufficiently approximates the original model (2).

**Example 3.3.** Consider the same setting in Example 3.1. Let $p_{c,x}(y) = (c/\pi)^{d/2}\mathrm{e}^{-c\|y-x\|^2}$ for $c > 0$ and $x \in \mathcal{X}_0$. Then, $p_{c,x} \in L^2(\mathbb{R}^d)$ and $\|p_{c,x}\|^2 = (2c/\pi)^{d/2}$. Since $p_{c,x}$ goes to the Dirac delta function centered at $x$ as $c \to \infty$, we have

$$F_c(g_1, \ldots, g_L, x) = \int_{\mathbb{R}^d} \rho(g_1)A_1\rho(g_2)A_2 \cdots A_{L-1}\rho(g_L)v(y)p_{c,x}(y)\mathrm{d}y.$$

Note that for any $x \in \mathbb{R}^d$ and any $g_1, \ldots, g_L \in G$, $\lim_{c \to \infty} F_c(g_1, \ldots, g_L, x) = f(g_1, \ldots, g_L)(x)$. Thus, if $c$ is sufficiently large, $F_c(g_1, \ldots, g_L, x)$ approximates $f(g_1, \ldots, g_L)(x)$ well.

### 3.2 RKHS FOR ANALYZING DEEP MODELS

We use the Rademacher complexity to derive a generalization bound. According to Theorem 3.5 in Mohri et al. (2018), the generalization error is bounded by the Rademacher complexity. Thus, if we obtain a Rademacher complexity bound, then we can also bound the generalization error. To derive a Rademacher complexity bound, we apply the framework of RKHSs. The Hilbert space $\mathcal{H}$ to which the models belong does not always have the reproducing property. Indeed, a typical example of $\mathcal{H}$ is $L^2(\mathbb{R}^d)$ as we discussed in Example 3.1. Thus, we consider an RKHS that is a function space on the parameter space $G$ and isomorphic to a subspace of $\mathcal{H}$. We can regard the deep model on the data space $\mathcal{X}_0$ as a function on $G$ through this isomorphism and make use of the reproducing property on $G$. Here, the isomorphism ensures that the mathematical structure of the RKHS is the same as the subspace of $\mathcal{H}$. We define the following positive definite kernel $k : (G \times \cdots \times G) \times (G \times \cdots \times G) \to \mathbb{C}$ to construct an RKHS to analyze the deep model (2):

$$k((g_1, \ldots, g_L), (\tilde{g}_1, \ldots, \tilde{g}_L)) = \langle \rho(g_1)A_1 \cdots A_{L-1}\rho(g_L)v, \rho(\tilde{g}_1)A_1 \cdots A_{L-1}\rho(\tilde{g}_L)v \rangle_{\mathcal{H}}.$$

We denote the RKHS associated with $k$ as $\mathcal{R}_k$.

Let $\mathbf{g} = (g_1, \ldots, g_L)$, $\phi(\mathbf{g}) = k(\cdot, \mathbf{g})$, and $\tilde{\phi}(\mathbf{g}) = \rho(g_1)A_1 \cdots A_{L-1}\rho(g_L)v$. Let $\mathcal{K}_0 = \{\sum_{i=1}^n c_i\tilde{\phi}(\mathbf{g}_i) \mid n \in \mathbb{N}, \mathbf{g}_i \in G^L, c_i \in \mathbb{C}\}$ and $\mathcal{K} = \overline{\mathcal{K}_0}$. Note that $\mathcal{K}$ is a sub-Hilbert space of $\mathcal{H}$. Let $\iota : \mathcal{K} \to \mathcal{R}_k$ defined as $\iota(h) = (\mathbf{g} \mapsto \langle \tilde{\phi}(\mathbf{g}), h \rangle_{\mathcal{H}})$. The map $\iota$ enables us to regard the Hilbert space $\mathcal{K}$, where the deep model is defined, as the RKHS $\mathcal{R}_k$.

**Proposition 3.4.** *The map $\iota$ is isometrically isomorphic.*

If $\rho$ is irreducible and $A_1, \ldots, A_L$ are invertible, then we have $\mathcal{K} = \mathcal{H}$, which means that the deep model (2) has universality. The following lemmas are derived using Lemmas 2.7 and 2.8.

**Lemma 3.5.** *Assume $\rho$ is irreducible. Let $\mathcal{A} = \{\sum_{i=1}^n c_i\rho(g_i) \mid n \in \mathbb{N}, g_i \in G, c_i \in \mathbb{C}\}$. Then, $\mathcal{A}$ is dense in $B(\mathcal{H})$ with respect to the strong operator topology.*

**Lemma 3.6.** *Assume $\rho$ is irreducible and $A_1, \ldots, A_{L-1}$ are invertible. Then, $\mathcal{K} = \overline{\mathcal{K}_0} = \mathcal{H}$.*

## 4 RADEMACHER COMPLEXITY BOUND

We apply the isomorphism in Proposition 3.4 to derive a Rademacher complexity bound with the aid of the reproducing property in the RKHS $\mathcal{R}_k$. If $p_{c,x} \in \mathcal{K}$, Eq. (3) implies $F_c(\cdot, x) = \iota(p_{c,x}) \in \mathcal{R}_k$ for $x \in \mathcal{X}_0$ and $c > 0$. Thus, we can apply the reproducing property with respect to the model $F_c(\cdot, x)$. Let $\Omega$ be a probability space equipped with a probability measure $P$. Let $S \in \mathbb{N}$ be the sample size, $x_1, \ldots, x_S \in \mathcal{X}_0$, and $\epsilon_1, \ldots, \epsilon_S : \Omega \to \mathbb{C}$ be i.i.d. Rademacher variables (random variables following the uniform distribution on $\{-1, 1\}$). For a measurable function $\epsilon : \Omega \to \mathbb{C}$, we denote by $\mathrm{E}[\epsilon]$ the integral $\int_\Omega \epsilon(\omega)\mathrm{d}P(\omega)$. The empirical Rademacher complexity $\hat{R}(\mathcal{F}, x_1, \ldots, x_S)$ of a function class $\mathcal{F}$ is defined as $\hat{R}(\mathcal{F}, x_1, \ldots, x_S) = \mathrm{E}[\sup_{F \in \mathcal{F}} \sum_{s=1}^S F(x_s)\epsilon_s]/S$. We denote by $\mathcal{F}_c$ the function class $\{F_c(g_1, \ldots, g_L, \cdot) \mid g_1, \ldots, g_L \in G\}$. The Rademacher complexity of $\mathcal{F}_c$ is upper bounded as follows.

**Theorem 4.1.** *Assume $p_{c,x} \in \mathcal{K}$ for $x \in \mathcal{X}_0$. Then, the Rademacher complexity of the function class $\mathcal{F}_c$ is bounded as*

$$\hat{R}(\mathcal{F}_c, x_1, \ldots, x_S) \leq \frac{\|A_1\| \cdots \|A_{L-1}\| \|v\| E(c)}{\sqrt{S}}.$$

**Remark 4.2.** *If $\rho$ is irreducible and $A_1, \ldots, A_{L-1}$ are invertible, then by Lemma 3.6, the assumption of Theorem 4.1 is satisfied automatically.*

**Remark 4.3.** *If $p_{c,x}(y) = (c/\pi)^{d/2} e^{-c\|y-x\|^2}$, then we have $E(c) = (2c/\pi)^{d/2}$. Combining with the discussion in Example 3.3, we can see that there is a tradeoff between $F_c$ being close to the original model $f$ and the constant $E(c)$ becoming large.*

An important example of models that can be analyzed using this framework is invertible neural networks.

## 4.1 INVERTIBLE NEURAL NETWORKS

Consider the same setting in Example 3.1. Note that since $\rho$ is irreducible, the assumption of Theorem 4.1 is satisfied in this case (see Remark 4.2). Let $nn(\mathbf{g}, x) = v(W_L \sigma_{L-1}(W_{L-1} \cdots \sigma_1(W_1 x - W_1 b_1) \cdots - W_{L-1}b_{L-1}) - W_L b_L)$ be a neural network model. Then, we have $f(\mathbf{g}) = nn(\mathbf{g}, \cdot) |\det W_1|^{1/2} \cdots |\det W_L|^{1/2}$. Thus, we have $F_c(\mathbf{g}, \cdot) = NN_c(\mathbf{g}, \cdot) |\det W_1|^{1/2} \cdots |\det W_L|^{1/2}$, where $NN_c(\mathbf{g}, x) = \int_{\mathbb{R}^d} nn(y) p_{c,x}(y) \mathrm{d}y$. Let $D > 0$ and $\mathcal{NN}_c = \{NN_c(\mathbf{g}, \cdot) \mid \mathbf{g} \in G^L, |\det W_1|^{-1/2}, \ldots, |\det W_L|^{-1/2} \leq D\}$. We assume $A_l$ is invertible for $l = 1, \ldots, L-1$.

**Theorem 4.4.** *The Rademacher complexity bound of $\mathcal{NN}_c$ is*

$$\hat{R}(\mathcal{NN}_c, x_1, \ldots, x_S) \leq \frac{E(c)\|v\| \prod_{l=1}^{L-1} \|A_l\|}{\sqrt{S}} \sup_{|\det W_l|^{-1/2} \leq D} \prod_{l=1}^{L} |\det W_l|^{-1/2}.$$

For example, if $\sigma_l$ is the elementwise Leaky ReLU, then $\|A_l\|$ is bounded as Lemma 2.5. Since $\det W_l$ is the product of the singular values of $W_l$, and it is in the denominator of the bound, Theorem 4.1 implies that the model can generalize well even if $W_l$ has large singular values.

# 5 GENERALIZATION TO NON-CONSTANT WIDTH NEURAL NETWORKS

## 5.1 ALGEBRAIC REPRESENTATION OF DEEP MODELS WITH KOOPMAN OPERATORS

In pervious sections, we focused on a single Hilbert space $\mathcal{H}$ and consider operators on $\mathcal{H}$. This corresponds to considering a neural network with a constant width. In addition, $\mathcal{H}$ is determined by the group representation, which forces us to consider a certain data space such as $\mathbb{R}^d$. However, in general, the width is not always constant. In addition, the data space is bounded in many cases. To meet this situation, we consider multiple Hilbert spaces $\mathcal{H}_0, \ldots, \mathcal{H}_{L-1}, \tilde{\mathcal{H}}_1, \ldots, \tilde{\mathcal{H}}_L$. Let $\Theta_l$ be a set of parameters and let $\eta_l : \Theta_l \to B(\tilde{\mathcal{H}}_l, \mathcal{H}_{l-1})$ for $l = 1, \ldots, L$. In addition, let $v \in \tilde{\mathcal{H}}_L$ and $A_l \in B(\mathcal{H}_l, \tilde{\mathcal{H}}_l)$ be fixed. Consider the model

$$f(\theta_1, \ldots, \theta_L) = \eta_1(\theta_1) A_1 \eta_2(\theta_2) \cdots A_{L-1} \eta_L(\theta_L) v,$$

where $\theta_l \in \Theta_l$ for $l = 1, \ldots, L$.

In the same manner as Subsection 3.1, we consider a regularized model

$$F_c(\theta_1, \ldots, \theta_L, x) = \langle \eta_1(\theta_1) A_1 \eta_2(\theta_2) \cdots A_{L-1} \eta_L(\theta_L) v, p_{c,x} \rangle_{\mathcal{H}_0},$$

where $p_{c,x} \in \mathcal{H}_0$ for $x \in \mathcal{X}_0$ and $c > 0$ with $\|p_{c,x}\| \leq E(c)$ for $E(c) > 0$. We also define a positive definite kernel $k : (\Theta_1 \times \cdots \times \Theta_L) \times (\Theta_1 \times \cdots \times \Theta_L) \to \mathbb{C}$ to construct an RKHS to analyze the deep model (2):

$$k((\theta_1, \ldots, \theta_L), (\tilde{\theta}_1, \ldots, \tilde{\theta}_L)) = \langle \eta_1(\theta_1) A_1 \cdots A_{L-1} \eta_L(\theta_L) v, \eta_1(\tilde{\theta}_1) A_1 \cdots A_{L-1} \eta_L(\tilde{\theta}_L) v \rangle_{\mathcal{H}_0}.$$

We set $\mathcal{R}_k$ and $\mathcal{K}$ in the same manner as in Subsection 3.2. This generalization allows us to derive Rademacher complexity bounds for a wide range of models.

## 5.2 NEURAL NETWORK WITH INJECTIVE WEIGHT MATRICES

Let $d_l \in \mathbb{N}$ and $\Theta_l = \{W \in \mathbb{C}^{d_l \times d_{l-1}} \mid W \text{ is injective}\}$. Let $\mathcal{X}_0 \subset \mathbb{R}^{d_0}$, $W_l \sigma_{l-1}(W_{l-1} \cdots W_2 \sigma_1(W_1 \mathcal{X}_0)) \subseteq \tilde{\mathcal{X}}_l \subseteq \mathbb{R}^{d_l}$, and $\sigma_l(W_l \cdots W_2 \sigma_1(W_1 \mathcal{X}_0)) \subseteq \mathcal{X}_l \subseteq \mathbb{R}^{d_l}$ that satisfy $\mu_{\mathbb{R}^{d_l}}(\mathcal{X}_l) > 0$ and $\mu_{\mathbb{R}^{d_l}}(\tilde{\mathcal{X}}_l) > 0$.

Starting from $\mathcal{X}_0$, we recurrently construct $\tilde{\mathcal{X}}_l$ and $\mathcal{X}_l$ for $l = 1, \ldots, L$. Since $W_l$ is injective, the space $W_l \mathcal{X}_{l-1}$ is $d_{l-1}$-dimensional. If $d_l > d_{l-1}$, then the measure $\mu_{\mathbb{R}^{d_l}}(W_l \mathcal{X}_{l-1})$ becomes 0, and setting $\tilde{\mathcal{X}}_l = W_l \mathcal{X}_{l-1}$ makes the analysis meaningless. Thus, we set a

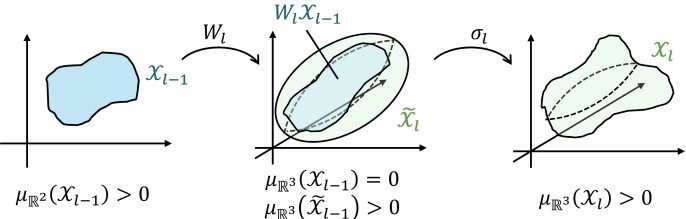

Figure 2: Construction of $\mathcal{X}_l$ and $\tilde{X}_l$

space $\mathcal{X}_l$ that includes $W_l \mathcal{X}_{l-1}$ and $\mu_{\mathbb{R}^{d_l}}(W_l \mathcal{X}_{l-1}) > 0$. Figure 2 schematically shows the construction of $\mathcal{X}_l$ and $\tilde{X}_l$. Let $\tilde{\mathcal{H}}_l = L^2(\tilde{\mathcal{X}}_l)$, $\mathcal{H}_l = L^2(\mathcal{X}_l)$, and $\eta_l(W_l) = K_{W_l}$ be the Koopman operator from $\tilde{\mathcal{H}}_l$ to $\mathcal{H}_{l-1}$ with respect to $W_l$. In addition, let $A_l = K_{\sigma_l}$ be the Koopman operator from $\mathcal{H}_l$ to $\tilde{\mathcal{H}}_l$ with respect to an activation function $\sigma_l : \tilde{\mathcal{X}}_l \to \mathcal{X}_l$ that satisfies Assumption 2.2. Then, we have

$$f(W_1, \ldots, W_L)(x) = v(W_L \sigma_{L-1}(W_{L-1}\sigma_{L-2}(\cdots \sigma_1(W_1 x)))).$$

Let $D > 0$ and $\mathcal{F}_c = \{F_c(\theta_1, \ldots, \theta_L, \cdot) \mid |\det W_1^* W_1|^{-1/4}, \ldots, |\det W_L^* W_L|^{-1/4} \leq D\}$. Let $\alpha(h) = (\int_{W_l \mathcal{X}_{l-1}} |h(x)|^2 d\mu_{\mathcal{R}(W_l)}(x) / \int_{\tilde{\mathcal{X}}_l} |h(x)|^2 d\mu_{\mathbb{R}^{d_l}}(x))^{1/2}$ for $h \in \tilde{\mathcal{H}}_l$. This value depends on how large we set $\tilde{\mathcal{X}}_l$ compared with $W_l \mathcal{X}_{l-1}$, and by setting $\tilde{\mathcal{X}}_l$ sufficiently large, we can bound it by 1 with a reasonable assumption (see Remark 5.3 for more details). In the same way as in Theorem 4.1, we obtain the following bound.

**Theorem 5.1.** *Assume $p_{c,x} \in \mathcal{K}$ for $x \in \mathcal{X}_0$. Let $f_l = v \circ W_L \circ \sigma_{L-1} \circ \cdots \circ W_{l+1} \circ \sigma_l$. Then, we have*

$$\hat{R}(\mathcal{F}_c, x_1, \ldots, x_S) \leq \sup_{|\det W_l^* W_l|^{-1/4} \leq D} \frac{E(c)\|v\| \prod_{l=1}^{L-1} \|A_l\| \alpha(f_l)}{\sqrt{S} \prod_{l=1}^L |\det W_l^* W_l|^{1/4}}, \tag{4}$$

As for $\|A_l\|$, since $A_l = K_{\sigma_l}$, we can evaluate the upper bound of $\|A_l\|$ by Lemma 2.3. For example, if $\mathcal{X}_0$ is bounded, we can apply Lemma 2.4 to the sigmoid and hyperbolic tangent.

**Remark 5.2.** *For simplicity, we consider models without bias terms. We obtain the same result for models with bias terms since the norm of the Koopman operator with respect to the shift function is 1.*

**Remark 5.3.** *Assume there exist $a, b > 0$ such that $a \leq |f_l(x)|^2 \leq b$. We set $\tilde{\mathcal{X}}_l$ sufficiently large so that $b \cdot \mu_{\mathcal{R}(W_l)}(W_l \mathcal{X}_{l-1}) \leq a \cdot \mu_{\mathbb{R}^{d_l}}(\tilde{\mathcal{X}}_l)$. Then, we have*

$$\alpha(f_l)^2 = \frac{\int_{W_l \mathcal{X}_{l-1}} |f_l(x)|^2 d\mu_{\mathcal{R}(W_l)}(x)}{\int_{\tilde{\mathcal{X}}_l} |f_l(x)|^2 d\mu_{\mathbb{R}^{d_l}}(x)} \leq \frac{b \cdot \mu_{\mathcal{R}(W_l)}(W_l \mathcal{X}_{l-1})}{a \cdot \mu_{\mathbb{R}^{d_l}}(\tilde{\mathcal{X}}_l)} \leq 1.$$

**Remark 5.4.** *There is a tradeoff between the magnitudes of the denominator and the numerator of the bound (4). When $\sigma_l(x)$ tends to be constant as $\|x\| \to \infty$, such as the hyperbolic tangent and sigmoid, the derivative of $\sigma_l^{-1}(x)$ tends to be large as the magnitude of $\|x\|$ becomes large. In this case, according to Lemma 2.3, if $\det W_l$ is large, then $\|A_l\|$ is also large since the volume of $\mathcal{X}_l$ becomes large. The activation function plays a significant role in increasing the complexity in this case. When $\sigma_1, \ldots, \sigma_{L-1}$ are unbounded, such as the Leaky ReLU, $\tilde{\mathcal{X}}_L$ becomes large if $\det W_1, \ldots, \det W_L$ are large, which makes $\|v\|$ large. The final nonlinear transformation $v$ plays a significant role in increasing the complexity in this case.*

**Advantage over existing Koopman-based bounds**  Hashimoto et al. (2024) proposed Rademacher complexity bounds using Koopman operator norms. Since the norm is defined by the Sobolev space,

the framework accepts only smooth and unbounded activation functions. In addition, although they include factors of the norms of Koopman operators with respect to the activation functions, their evaluation is extremely challenging, making the effect of the activation function unclear. On the other hand, our bound can be applied to various types of activation functions, such as the hyperbolic tangent, sigmoid, and Leaky ReLU, we can evaluate the Koopman operator norms using Lemmas 2.3 – 2.5, and we can understand the effect of the activation function as discussed in Remark 5.4.

## 5.3 GENERAL NEURAL NETWORK

If $W$ is not injective, the Koopman operator $K_W$ is unbounded. Thus, instead of the standard Koopman operators, we consider weighted Koopman operators. Let $d_l \in \mathbb{N}$ and $\Theta_l = \{W \in \mathbb{C}^{d_l \times d_{l-1}}\}$. For $l = 0, \ldots, L - 1$, let $\tilde{d}_l = \dim(\ker(W_{l+1}))$, $q_1, \ldots, q_{\tilde{d}_l}$ be an orthonormal basis of $\ker(W_{l+1})$, $q_{\tilde{d}_l+1}, \ldots, q_{d_l}$ be an orthonormal basis of $\ker(W_{l+1})^\perp$, $\mathcal{X}_l = \{\sum_{i=1}^{d_l} c_i q_i \mid c_i \in [a_i, b_i]\}$ for some $a_i < b_i$ such that $\sigma_l(W_l \cdots W_2 \sigma_1(W_1 \mathcal{X}_0)) \subseteq \mathcal{X}_l$, $\mathcal{Y}_l = \{\sum_{i=1}^{\tilde{d}_l} c_i q_i \mid c_i \in [a_i, b_i]\}$, and $\mathcal{Z}_l = \{\sum_{i=\tilde{d}_l+1}^{d_l} c_i q_i \mid c_i \in [a_i, b_i]\}$. Let $W_l \sigma_{l-1}(W_{l-1} \cdots W_2 \sigma_1(W_1 \mathcal{X}_0)) \subseteq \tilde{\mathcal{X}}_l \subseteq \mathbb{R}^{d_l}$ satisfying $\mu_{\mathbb{R}^{d_l}}(\tilde{\mathcal{X}}_l) > 0$.

In this case, to decompose the integral on $\ker(W_{l+1})$ and that on $\ker(W_{l+1})^\perp$, we set the orthonormal basis along $\ker(W_{l+1})$ and define $\mathcal{Y}_l$ and $\mathcal{Z}_l$. Figure 3 schematically shows the construction of $\mathcal{X}_l$, $\mathcal{Y}_l$, $\mathcal{Z}_l$, and $\tilde{X}_l$. Let $\tilde{\mathcal{H}}_l$ and $\mathcal{H}_l$ be the same space as in Subsection 5.2, and Let

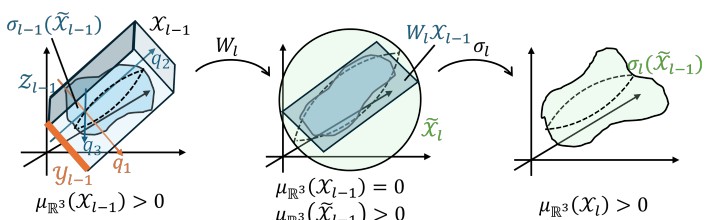

Figure 3: Construction of $\mathcal{X}_l$, $\mathcal{Y}_l$, $\mathcal{Z}_l$, and $\tilde{X}_l$

$\eta_l(W) = \hat{K}_{\psi_l, W}$ be the weighted Koopman operator from $\tilde{\mathcal{H}}_l$ to $\mathcal{H}_{l-1}$ with respect to $W$ and $\psi_l$, where $\psi_l$ is defined as $\psi_l(x) = \psi_l(x_1) = 1$ for $x \in \mathcal{X}_{l-1}$, where $x = x_1 + x_2$ with $x_1 \in \mathcal{Y}_{l-1}$ and $x_2 \in \mathcal{Z}_{l-1}$, and $\psi_l(x) = 0$ for $x \notin \mathcal{X}_{l-1}$. In addition, let $A_l$ be the same operator as in Subsection 5.2. Then, we have

$$f(W_1, \ldots, W_L)(x) = \psi_1(x)\psi_2(\sigma_1(W_1 x)) \cdots \psi_L(\sigma_{L-1}(W_{L-1}\sigma_{L-2}(\cdots \sigma_1(W_1 x)))) \\ \cdot v(W_L \sigma_{L-1}(W_{L-1}\sigma_{L-2}(\cdots \sigma_1(W_1 x)))).$$

The factor $\psi_1(x) \cdots \psi_L(\sigma_{L-1}(W_{L-1}\sigma_{L-2}(\cdots \sigma_1(W_1 x))))$ is an auxiliary factor. Since $\psi_l(x) = 1$ for $x \in \mathcal{X}_{l-1}$, we have $f(W_1, \ldots, W_L)(x) = v(W_L \sigma_{L-1}(W_{L-1}\sigma_{L-2}(\cdots \sigma_1(W_1 x))))$ for $x \in \mathcal{X}_0$ in the data space, exactly the same structure as that of neural networks. Thus, we can regard $f$ as the original neural network $v(W_L \sigma_{L-1}(W_{L-1}\sigma_{L-2}(\cdots \sigma_1(W_1 x))))$.

Let $D > 0$ and $\mathcal{F}_c = \{F_c(\theta_1, \ldots, \theta_L, \cdot) \mid |\det W_1|_{\ker(W_1)^\perp}|^{-1/2}, \ldots, |\det W_L|_{\ker(W_L)^\perp}|^{-1/2} \leq D\}$. In the same way as in Theorem 5.1, we obtain the following bound.

**Theorem 5.5.** *Assume $p_{c,x} \in \mathcal{K}$ for $x \in \mathcal{X}_0$. Then, we have*

$$\hat{R}(\mathcal{F}_c, x_1, \ldots, x_S) \leq \sup_{|\det W_l|_{\ker(W_l)^\perp}|^{-1/2} \leq D} \frac{E(c)\|v\| \prod_{l=1}^{L-1} \|A_l\| \alpha(f_l) \prod_{l=1}^{L} \mu_{\ker(W_l)}(\mathcal{Y}_{l-1})}{\sqrt{S} \prod_{l=1}^{L} |\det W_l|_{\ker(W_l)^\perp}|^{1/2}}.$$

**Remark 5.6.** *If the output of the $l$th layer has small values in the direction of $\ker(W_{l+1})$, then the factor $\mu_{\ker(W_{l+1})}(\mathcal{Y}_l)$ is small. We expect that the magnitude of the noise is smaller than that of the essential signals. This implies that if the weight $W_{l+1}$ is learned so that $\ker(W_{l+1})$ becomes the direction of noise, i.e., so that the noise is removed by $W_{l+1}$, the model generalizes well. Arora et al. (2018) insist that the noise stability property implies that the model generalizes well. The result of Theorem 5.5 does not contradict the results of Arora et al. (2018).*

## 5.4 CONVOLUTIONAL NEURAL NETWORK

Let $I_l = J_{l,1} \times \cdots \times J_{l,d_l} \subseteq \mathbb{Z}^{d_l}$ be a finite index set and $\Theta_l = \{\theta \in \mathbb{R}^{I_l} \mid x \mapsto \theta * x \text{ is invertible}\}$. Let $\theta_l \in \Theta_l$, $P_l$ be the matrix representing the average pooling with pool size $m_l$, which is defined

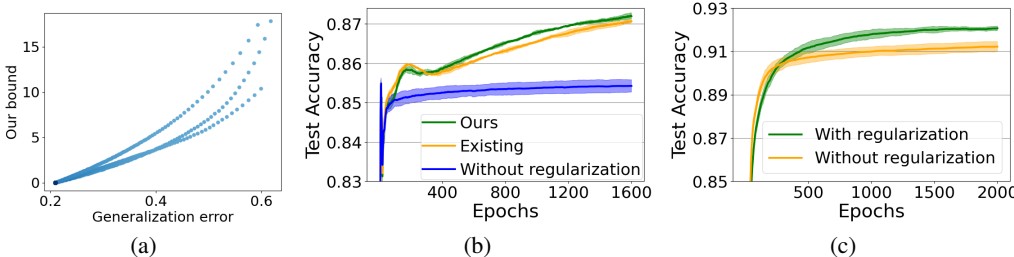

Figure 4: (a) Scatter plot of the generalization error versus our bound (for 3 independent runs). The color is set to get dark as the epoch proceeds. (b) Test accuracy with the regularization based on our bound and that based on the existing bound (deep neural net with dense layers). (c) Test accuracy with and without the regularization based on our bound (LeNet).

as $(P_l)_{i,j} = 1/m_l$ for $i, j \in I_l$ if the $j$th element of the input is pooled in the $i$th element of the output, and $\sigma_l$ be the same as in Subsection 5.2. Let $\mathcal{X}_0 \subseteq \mathbb{R}^{I_1}$, $\theta_l * P_{l-1}\sigma_{l-1}(\theta_{l-1} * \cdots \theta_2 * P_1\sigma_1(\theta_1 * \mathcal{X}_0)) \subseteq \tilde{\mathcal{X}}_l \subseteq \mathbb{R}^{I_l}$ and $P_l\sigma_l(\theta_l * \cdots \theta_2 * P_1\sigma_1(\theta_1 * \mathcal{X}_0)) \subseteq \mathcal{X}_l \subseteq \mathbb{R}^{I_{l+1}}$ satisfying $\mu_{\mathbb{R}^{I_l}}(\tilde{\mathcal{X}}_l) > 0$ and $\mu_{\mathbb{R}^{I_{l+1}}}(\mathcal{X}_l) > 0$. Let $\tilde{d}_l = \dim(\ker(P_l))$, $q_1, \ldots, q_{\tilde{d}_l}$ be an orthonormal basis of $\ker(P_l)$, $q_{\tilde{d}_l+1}, \ldots, q_{d_l}$ be an orthonormal basis of $\ker(P_l)^\perp$, $\hat{\mathcal{X}}_l = \{\sum_{i=1}^{d_l} c_i q_i \mid c_i \in [a_i, b_i]\}$ for some $a_i < b_i$ such that $\sigma_l(\theta_l * \cdots \theta_2 * P_1\sigma_1(\theta_1 * \mathcal{X}_0)) \subseteq \hat{\mathcal{X}}_l \subseteq \mathbb{R}^{I_l}$, $\hat{\mathcal{Y}}_l = \{\sum_{i=1}^{\tilde{d}_l} c_i q_i \mid c_i \in [a_i, b_i]\}$, and $\hat{\mathcal{Z}}_l = \{\sum_{i=\tilde{d}_l+1}^{d_l} c_i q_i \mid c_i \in [a_i, b_i]\}$. Let $\mathcal{H}_l = L^2(\mathcal{X}_l)$, $\tilde{\mathcal{H}}_l = L^2(\tilde{\mathcal{X}}_l)$, $\hat{\mathcal{H}}_l = L^2(\hat{\mathcal{X}}_l)$, and $\eta_l : \Theta_l \to B(\tilde{\mathcal{H}}_l, \mathcal{H}_{l-1})$ be defined as $\eta_l(\theta)h(x) = h(\theta * x)$, where $*$ is the convolution. Note that the convolution is a linear operator whose eigenvalues are Fourier components $\gamma_m(\theta_l) := \sum_{j \in I_l} \theta_j e^{i(S_l\, j) \cdot m}$ for $m \in I_l$, where $S_l$ is the diagonal matrix whose diagonal is the scaling factor $[1/(2\pi|J_{l,1}|), \ldots, 1/(2\pi|J_{l,d_l}|)]$. Let $A_l = K_{\sigma_l}\tilde{K}_{\psi_l, P_l}$, where $\tilde{K}_{\psi_l, P_l}$ and $K_{\sigma_l}$ are weighted Koopman and Koopman operators from $\mathcal{H}_l$ to $\hat{\mathcal{H}}_l$ and from $\hat{\mathcal{H}}_l$ to $\tilde{\mathcal{H}}_l$, respectively. Here, $\psi_l$ is defined as $\psi_l(x) = \psi_l(x_1) = 1$ for $x \in \hat{\mathcal{X}}_l$, where $x = x_1 + x_2$ with $x_1 \in \hat{\mathcal{Y}}_l$ and $x_2 \in \hat{\mathcal{Z}}_l$, and $\psi_l(x) = 0$ for $x \notin \hat{\mathcal{X}}_l$. Then, we have

$$f(\theta_1, \ldots, \theta_L)(x) = \psi_1(\sigma_1(\theta_1 * x)) \cdots \psi_{L-1}(\sigma_{L-1}(\theta_{L-1} * P_{L-2}\sigma_{L-2}(\cdots P_1\sigma_1(\theta_1 * x))))$$
$$\cdot v(\theta_L * P_{L-1}\sigma_{L-1}(\theta_{L-1} * \cdots * P_1\sigma_1(\theta_1 * x) \cdots)).$$

Let $\beta_l(\theta) = \prod_{m \in I_l} \gamma_m(\theta)$ and $\mathcal{F}_c = \{F_c(\theta_1, \ldots, \theta_L, \cdot) \mid |\beta(\theta_1)|^{-1/2}, \ldots, |\beta(\theta_L)|^{-1/2} \leq D\}$.

**Proposition 5.7.** *Assume $p_{c,x} \in \mathcal{K}$ for $x \in \mathcal{X}_0$. Then, we have*

$$\hat{R}(\mathcal{F}_c, x_1, \ldots, x_S) \leq \sup_{|\beta(\theta_l)|^{-1/2} \leq D} \frac{E(c)\|v\| \prod_{l=1}^{L-1} \|A_l\| \mu_{\ker(P_l)}(\hat{\mathcal{Y}}_l)}{\sqrt{S} \prod_{l=1}^{L} |\beta_l(\theta_l)|^{1/2}}.$$

**Remark 5.8.** *If $\sigma_l$ is bounded, then we can set $\hat{X}_l$ independent of $\theta_1, \ldots, \theta_l$ so that it covers the range of $\sigma_l$. Since $P_l$ is a fixed operator, the factor $\mu_{\ker(P_1)}(\hat{\mathcal{Y}}_l)$ is a constant in this case.*

## 6  NUMERICAL RESULTS

We numerically confirm the validity of the proposed bound. Experimental details are in Appendix B.

**Validity of the bound**  To show the relationship between the generalization error and the proposed bound, we consider a regression problem with synthetic data on $\mathcal{X}_0 = [-1, 1]^3$. The target function $t$ is $t(x) = e^{-\|2x-1\|^2}$. We constructed a network $f(x) = v(W_2\sigma(W_1 x + b_1) + b_2)$, where $W_1 \in \mathbb{R}^{3 \times 3}$, $W_2 \in \mathbb{R}^{6 \times 3}$, $b_1 \in \mathbb{R}^3$, $b_2 \in \mathbb{R}^6$, $v(x) = w_3 e^{-\|x\|^2}$, $w_3 \in \mathbb{R}$, and $\sigma$ is the elementwise hyperbolic tangent. We created a training dataset from randomly drawn samples from the uniform distribution on $[-1, 1]^3$. The training sample size $S$ is 1000. Our bound is proportional to the value $r := |w_3| \sup_{[x_1, x_2, x_3] \in \sigma(W_1\mathcal{X}_0 + b_1)} 1/(1 - x_1^2)/(1 - x_2^2)/(1 - x_3^2)|\det W_1^* W_1|^{-1/4} \cdot |\det W_2^* W_2|^{-1/4}$ since $\|v\| \leq |w_3| \int_{\mathbb{R}^6} e^{-\|x\|} dx$ and according to Lemma 2.4. We added $0.1r$ as a regularization term. Figure 4 (a) illustrates the relationship between the generalization error and our bound throughout the learning process. We can see that the generalization bound gets small in proportion to our bound.

**Comparison with existing bounds**  To compare our bound with existing bounds, we considered the same classification task with MNIST as in Hashimoto et al. (2024). We constructed the same model $f(x) = \sigma_4(W_4\sigma(W_3\sigma(W_2\sigma(W_1x + b_1) + b_2) + b_3) + b_4)$ as Hashimoto et al. (2024) with dense layers. Based on the bound, we tried to make the factors $\|A_l\|$, $1/\det W_l^* W_l^{1/2}$, and $\|v\|$ small, where $v(x) = \sigma_4(W_4\sigma(W_3x + b_3) + b_4)$, $\sigma(x_1, \ldots, x_d) = [\tilde{\sigma}(x_1), \ldots, \tilde{\sigma}(x_d)]$ is the elementwise smooth Leaky ReLU proposed by Biswas et al. (2022), and $\sigma_4$ is the softmax. This setting is for meeting the setting in (Hashimoto et al., 2024). We set $\mathcal{X}_0 = [0, 1]^{784}$, $\tilde{\mathcal{X}}_1 = (\|W_1\| + \|b_1\|_\infty)[-1, 1]^{1024} \supseteq W_1\mathcal{X}_0 + b_1$, $\mathcal{X}_1 = \sigma(\tilde{\mathcal{X}}_1) \supseteq \sigma(W_1\mathcal{X}_0 + b_1)$, $\tilde{\mathcal{X}}_2 = (\|W_2\|(\|W_1\| + \|b_1\|_\infty) + \|b_2\|_\infty)[-1, 1]^{2048}$, and $\mathcal{X}_2 = \sigma(\tilde{\mathcal{X}}_2) \supseteq \sigma(W_2\sigma(W_1\mathcal{X}_0 + b_1) + b_2)$. To make the factor $\|A_l\|$ small, we applied Lemma 2.3 and set a regularization term $r_1 = \sup_{x\in(\mathcal{X}_1)_1} |(\tilde{\sigma}^{-1})'(x)| + \sup_{x\in(\mathcal{X}_2)_1} |(\tilde{\sigma}^{-1})'(x)|$. Here, $(\mathcal{X}_1)_1$ is the set of the first elements of the vectors in $\mathcal{X}_1$. In addition, we set $r_2 = 1/(1 + \det W_1^* W_1^{1/4}) + 1/(1 + \det W_2^* W_2^{1/4})$. Regarding $\|v\|$, we set $r_3 = \|W_1\| + \|W_2\|$ since we have $\|v\|^2 = \int_{\mathcal{X}_2} |v(x)|^2 \mathrm{d}x \leq \mu(\mathcal{X}_2) \leq \mu(\tilde{\mathcal{X}}_2)$. We added the regularization term $0.01(r_1 + r_2 + r_3)$ to the loss function. The training sample size is $S = 1000$. We compared the regularization based on our bound with that based on the bound proposed by Hashimoto et al. (2024). The result is shown in Figure 4 (b). Note that since the training sample size $S$ is small, obtaining a high test accuracy is challenging. We can see that with the regularization based on our bound, we obtain a better performance than that based on the existing bound.

**Validity for existing CNN models (LeNet)**  To show that our bound is valid for practical models, we applied the regularization based on our bound to LeNet on MNIST (Lecun et al., 1998). We set the activation function $\sigma$ of each layer as the elementwise hyperbolic tangent function and the final nonlinear transformation $v$ as the softmax. We used the same training and test datasets as the previous experiment. In addition, we set $\mathcal{X}_0 = [0, 1]^{784}$, $\tilde{\mathcal{X}}_l = (\|W_l\| + \|b_l\|_\infty)[-1, 1]^{1024} \supseteq W_l\sigma(\cdots\sigma(W_1\mathcal{X}_0 + b_1) + \cdots) + b_l$, $\mathcal{X}_l = \sigma(\tilde{\mathcal{X}}_l) \supseteq \sigma(W_l\sigma(\cdots\sigma(W_1\mathcal{X}_0 + b_1) + \cdots) + b_l)$. Here, $W_l$ is the matrix that represents the $l$th convolution layer. We note that the bound by Hashimoto et al. (2024) is not valid for the models with hyperbolic tangent and softmax functions. To make the factor $\|A_l\|$ small, we applied Lemmas 2.3 and 2.4 and tried to make $\inf_{x\in\mathcal{X}_l}(1 - x^2)$ large. Thus, we set a regularization term $r_1 = \sum_{l=1}^4 \sup_{x\in(\mathcal{X}_l)_1} 1/(1 + 1 - x^2)$. Regarding the factor $\det W_l |_{\ker(W_l)^\perp}^{-1/2}$, we set $r_2 = \|(0.01I + W_l W_l^*)^{-1}\| = 1/(0.01 + s_{\min}(W_l))$, to make $s_{\min}(W_l)$ large, where $s_{\min}(W_l)$ is the smallest singular value of $W_l$ since the determinant is described as the product of the singular values. For $\|v\|$, we set $r_3 = \|W_L\|$ in the same way as in the previous experiment according to the definition of $\tilde{\mathcal{X}}_L$. We added the regularization term $0.1(r_1 + r_2 + r_3)$ to the loss function and compared it with the case without regularization. The result is shown in Figure 4 (c). We can see that with the regularization, the model performs better than in the case without the regularization, which shows the validity of our bound for LeNet.

## 7    CONCLUSION AND LIMITATION

In this paper, we derived a new Koopman-based Rademacher complexity bound. Analogous to the existing Koopman-based bounds, our bound describes that neural networks with high-rank weight matrices can generalize well. Existing Koopman-based bounds rely on the smoothness of the function space and the unboundedness of the data space, which makes the result valid for limited neural network models with smooth and unbounded activation functions. We resolved this issue by introducing an algebraic representation of neural network models and constructing an RKHS associated with a kernel defined with this representation. Our bound is valid for a wide range of models, such as those with the hyperbolic tangent, sigmoid, and Leaky ReLU activation functions. Our framework is the first step to filling the gap between the Koopman-based analysis of generalization bounds and practical situations.

Although our bound can be applied to models more realistic than the existing Koopman-based bounds, it is not valid for activation functions whose derivative is zero in some domain, such as the exact ReLU. Introducing a variant of the Koopman operator such as the weighted Koopman operator may help us deal with this situation, but more detailed investigation is left for future work.

ACKNOWLEDGMENT

This work was supported by JSPS KAKENHI 24K21316, 25H01453, JST BOOST JPMJBY24E2, and JST CREST JPMJCR2015, JPMJCR25I5, JPMJCR24Q6, JPMJCR24Q1.

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

APPENDIX

A    PROOFS

We show the proofs of the theorems, propositions, and lemmas in the main text.

**Lemma 2.3**  *Assume $\sigma : \tilde{\mathcal{X}} \to \mathcal{X}$ is bijective, $\sigma^{-1}$ is differentiable, and the Jacobian of $\sigma^{-1}$ is bounded in $\mathcal{X}$. Then, we have $\|K_\sigma\| \leq \sup_{x \in \mathcal{X}} |J\sigma^{-1}(x)|^{1/2}$, where $J\sigma^{-1}$ is the Jacobian of $\sigma^{-1}$. In particular, the Koopman operator $K_\sigma$ is bounded.*

*Proof.* For $h \in L^2(\mathcal{X})$, we have

$$\|K_\sigma h\|^2 = \int_{\tilde{\mathcal{X}}} |h(\sigma(x))|^2 \mathrm{d}x = \int_{\mathcal{X}} |h(x)|^2 |J\sigma^{-1}(x)| \mathrm{d}x$$

$$\leq \sup_{x \in \mathcal{X}} |J\sigma^{-1}(x)| \int_{\mathcal{X}} |h(x)|^2 \mathrm{d}x = \sup_{x \in \mathcal{X}} |J\sigma^{-1}(x)| \|h\|^2.$$

$\square$

**Lemma 2.5**  *Let $\tilde{\mathcal{X}} = \mathcal{X} = \mathbb{R}^d$. Let $\sigma$ be the elementwise Leaky ReLU defined as $\tilde{\sigma}(x) = ax$ for $x \leq 0$ and $\tilde{\sigma}(x) = x$ for $x > 0$, where $a > 0$. Then, we have $\|K_\sigma\| \leq \max\{1, 1/a^d\}^{1/2}$.*

*Proof.* For $h \in L^2(\mathcal{X})$, we have

$$\|K_\sigma h\|^2 = \int_{\mathbb{R}^d} |h(\sigma(x))|^2 \mathrm{d}x$$

$$= \int_{(-\infty,0]^d} |h(ax)|^2 \mathrm{d}x + \int_{(0,\infty)\times(-\infty,0]^{d-1}} |h(\mathrm{diag}\{1, a, \ldots, a\}x)|^2 \mathrm{d}x + \cdots + \int_{(0,\infty)^d} |h(x)|^2 \mathrm{d}x$$

$$\leq \max\left\{1, 1/a^d\right\} \int_{\mathbb{R}^d} |h(x)|^2 \mathrm{d}x = \max\left\{1, 1/a^d\right\} \|h\|^2.$$

$\square$

**Proposition 3.4**  *The map $\iota$ is isometrically isomorphic.*

Proposition 3.4 is derived using the following lemmas.

**Lemma A.1.** *The map $\iota$ is injective.*

*Proof.* Assume $\iota(h) = 0$. Then, for any $\mathbf{g} \in G$, $\langle \tilde{\phi}(\mathbf{g}), h \rangle = 0$. Thus, for any $n \in \mathbb{N}$, $\mathbf{g}_1, \ldots, \mathbf{g}_n$, and $c_1, \ldots, c_n \in \mathbb{C}$, we have $\langle \sum_{i=1}^n c_i \tilde{\phi}(\mathbf{g}_i), h \rangle = 0$, which means for any $\tilde{h} \in \mathcal{K}_0$, $\langle \tilde{h}, h \rangle = 0$. Thus, we obtain $h = 0$. $\qquad\square$

**Lemma A.2.** *The map $\iota$ preserves the norm and is surjective.*

*Proof.* By definition, $\iota$ is a linear map that maps $\tilde{\phi}(\mathbf{g}) \in \mathcal{K}_0$ to $\phi(\mathbf{g}) \in \mathcal{R}_{k,0}$. Thus, we have $\iota(\mathcal{K}_0) = \mathcal{R}_{k,0}$.

For $h \in \mathcal{K}_0$, there exist $n \in \mathbb{N}$, $\mathbf{g}_1, \ldots, \mathbf{g}_n \in G^L$, and $c_1, \ldots, c_n \in \mathbb{C}$ such that $h = \sum_{i=1}^n c_i \tilde{\phi}(\mathbf{g}_i)$. We have

$$\|\iota(h)\|_{\mathcal{R}_k}^2 = \left\| \sum_{i=1}^n c_i \phi(\mathbf{g}_i) \right\|_{\mathcal{R}_k}^2 = \sum_{i,j=1}^n \overline{c_i} c_j k(\mathbf{g}_i, \mathbf{g}_j) = \sum_{i,j=1}^n \overline{c_i} c_j \langle \tilde{\phi}(\mathbf{g}_i), \tilde{\phi}(\mathbf{g}_j) \rangle_{\mathcal{H}} = \|h\|_{\mathcal{H}}^2.$$

Thus, $\iota$ preserves the norm, and in particular, it is bounded.

For any $r \in \mathcal{R}_k$, there exists a sequence $r_1, r_2, \ldots \in \mathcal{R}_{k,0}$ such that $\lim_{i \to \infty} r_i = r$. Since $\iota(\mathcal{K}_0) = \mathcal{R}_{k,0}$, there exists $h_i \in \mathcal{K}_0$ such that $\iota(h_i) = r_i$ for $i = 1, 2, \ldots$. Thus, we have $r = \lim_{i \to \infty} r_i = \lim_{i \to \infty} \iota(h_i) = \iota(\lim_{i \to \infty} h_i) = \iota(h)$. $\qquad\square$

**Lemma 3.5** *Assume $\rho$ is irreducible. Let $\mathcal{A} = \{ \sum_{i=1}^n c_i \rho(g_i) \mid n \in \mathbb{N}, g_i \in G, c_i \in \mathbb{C} \}$. Then, $\mathcal{A}$ is dense in $B(\mathcal{H})$ with respect to the strong operator topology.*

*Proof.* By the Schur's lemma (Lemma 2.7), the commutant of $\overline{\mathcal{A}}^{\mathrm{SOT}}$, the closure of $\mathcal{A}$ with respect to the strong operator topology, is $\mathbb{C}I$. Thus, the double commutant of $\overline{\mathcal{A}}^{\mathrm{SOT}}$ is $B(\mathcal{H})$. By the von Neumann double commutant theorem (Lemma 2.8), the double commutant of $\overline{\mathcal{A}}^{\mathrm{SOT}}$ is $\overline{\mathcal{A}}^{\mathrm{SOT}}$ itself. Therefore, we have $\overline{\mathcal{A}}^{\mathrm{SOT}} = B(\mathcal{H})$. $\qquad\square$

**Lemma 3.6** *Assume $\rho$ is irreducible and $A_1, \ldots, A_{L-1}$ are invertible. Then, $\mathcal{K} = \overline{\mathcal{K}_0} = \mathcal{H}$.*

*Proof.* Let $h \in \mathcal{H}$. Then, there exists $B \in B(\mathcal{H})$ such that $h = Bv$. Let $\varepsilon > 0$. By Lemma 3.5, there exist $n_L \in \mathbb{N}$, $g_{L,1}, \ldots, g_{L,n_L} \in G$, and $c_{L,1}, \ldots, c_{L,n_L} \in \mathbb{C}$ such that $\|\tilde{A}_L v - A_{L-1}^{-1} v\| \leq \varepsilon$, where $\tilde{A}_L = \sum_{\alpha_L=1}^{n_L} c_{L,\alpha_L} \rho(g_{L,\alpha_L})$. In addition, there exist $n_{L-1} \in \mathbb{N}$, $g_{L-1,1}, \ldots, g_{L-1,n_{L-1}} \in G$, and $c_{L-1,1}, \ldots, c_{L-1,n_{L-1}} \in \mathbb{C}$ such that $\|\tilde{A}_{L-1}(A_{L-1} \tilde{A}_L v) - A_{L-2}^{-1}(A_{L-1} \tilde{A}_L v)\| \leq \varepsilon$, where $\tilde{A}_{L-1} = \sum_{\alpha_{L-1}=1}^{n_{L-1}} c_{L-1,\alpha_{L-1}} \rho(g_{L-1,\alpha_{L-1}})$. We continue this process, and for $l = L-2, \ldots, 2$, we obtain $n_l \in \mathbb{N}$, $g_{l,1}, \ldots, g_{l,n_l} \in G$, and $c_{l,1}, \ldots, c_{l,n_l} \in \mathbb{C}$ such that $\|\tilde{A}_l(A_l \tilde{A}_{l+1} A_{l+1} \cdots \tilde{A}_{L-1} A_{L-1} \tilde{A}_L v) - A_{l-1}^{-1}(A_l \tilde{A}_{l+1} A_{l+1} \cdots \tilde{A}_{L-1} A_{L-1} \tilde{A}_L v)\| \leq \varepsilon$, where $\tilde{A}_l = \sum_{\alpha_l=1}^{n_l} c_{l,\alpha_l} \rho(g_{l,\alpha_l})$. Finally, we get $n_1 \in \mathbb{N}$, $g_{1,1}, \ldots, g_{1,n_1} \in G$, and $c_{1,1}, \ldots, c_{1,n_1} \in \mathbb{C}$ such that $\|\tilde{A}_1(A_1 \tilde{A}_2 A_2 \cdots \tilde{A}_{L-1} A_{L-1} \tilde{A}_L v) - B(A_1 \tilde{A}_2 A_2 \cdots \tilde{A}_{L-1} A_{L-1} \tilde{A}_L v)\| \leq \varepsilon$, where $\tilde{A}_1 = \sum_{\alpha_1=1}^{n_1} c_{1,\alpha_1} \rho(g_{1,\alpha_1})$. Let $C = \tilde{A}_1 A_1 \cdots \tilde{A}_{L-1} A_{L-1} \tilde{A}_L$. Then, we have

$$\|Cv - h\| \leq \|Cv - BA_1 \tilde{A}_2 \cdots A_{L-1} \tilde{A}_L v\| + \|BA_1 \tilde{A}_2 \cdots A_{L-1} \tilde{A}_L v - BA_2 \tilde{A}_3 \cdots A_{L-1} \tilde{A}_L v\|$$
$$+ \cdots + \|BA_{L-2} \tilde{A}_{L-1} A_{L-1} \tilde{A}_L v - BA_{L-1} \tilde{A}_L v\| + \|BA_{L-1} \tilde{A}_L v - B\tilde{A}_L v\|$$
$$\leq \varepsilon + \|BA_1\|\varepsilon + \cdots + \|BA_{L-2}\|\varepsilon + \|BA_{L-1}\|\varepsilon.$$

$\qquad\square$

**Theorem 4.1** *Let $\mathcal{F}_c$ the function class $\{F_c(g_1, \ldots, g_L, \cdot) \mid g_1, \ldots, g_L \in G\}$. Assume $p_{c,x} \in \mathcal{K}$ for $x \in \mathcal{X}_0$. Then, the Rademacher complexity of the function class $\mathcal{F}_c$ is bounded as*

$$\hat{R}(\mathcal{F}_c, x_1, \ldots, x_S) \leq \frac{\|A_1\| \cdots \|A_{L-1}\| \|v\| E(c)}{\sqrt{S}}.$$

*Proof.* Since $F_c(\cdot, x) = \iota(p_{c,x}) \in \mathcal{R}_k$, by the reproducing property, we have

$$\frac{1}{S}\mathrm{E}\left[\sup_{\mathbf{g} \in G^L} \sum_{s=1}^{S} F_c(\mathbf{g}, x_s)\epsilon_s\right] = \frac{1}{S}\mathrm{E}\left[\sup_{\mathbf{g} \in G^L} \left\langle \phi(\mathbf{g}), \sum_{s=1}^{S} F_c(\cdot, x_s)\epsilon_s \right\rangle_{\mathcal{R}_k}\right]$$

$$\leq \frac{1}{S} \sup_{\mathbf{g} \in G^L} \|\phi(\mathbf{g})\|_{\mathcal{R}_k} \mathrm{E}\left[\left\|\sum_{s=1}^{S} F_c(\cdot, x_s)\epsilon_s\right\|_{\mathcal{R}_k}\right]$$

$$= \frac{1}{S} \sup_{\mathbf{g} \in G^L} \|\tilde{\phi}(\mathbf{g})\|_{\mathcal{H}} \mathrm{E}\left[\left(\sum_{s,t=1}^{S} \langle F_c(\cdot, x_s)\epsilon_s, F_c(\cdot, x_t)\epsilon_t \rangle_{\mathcal{R}_k}\right)^{1/2}\right]$$

$$\leq \frac{1}{S} \sup_{\mathbf{g} \in G^L} \|\rho(g_1)A_1 \cdots A_{L-1}\rho(g_L)v\|_{\mathcal{H}} \left(\mathrm{E}\left[\sum_{s,t=1}^{S} \langle F_c(\cdot, x_s)\epsilon_s, F_c(\cdot, x_t)\epsilon_t \rangle_{\mathcal{R}_k}\right]\right)^{1/2}$$

$$\leq \frac{1}{S} \|A_1\| \cdots \|A_{L-1}\| \|v\| \left(\sum_{s=1}^{S} \|F_c(\cdot, x_s)\|_{\mathcal{R}_k}^2\right)^{1/2}, \tag{5}$$

where the third equality is by Lemma A.2, the fourth inequality is by the Jensen's inequality, and the final inequality is derived since $\rho(g_1) \ldots, \rho(g_L)$ are unitary.

Since $F_c(\cdot, x) = \iota(p_{c,x})$, we apply Lemma A.2 again and obtain

$$\frac{1}{S}\|A_1\| \cdots \|A_{L-1}\| \|v\| \left(\sum_{s=1}^{S} \|F_c(\cdot, x_s)\|_{\mathcal{R}_k}^2\right)^{1/2} = \frac{1}{S}\|A_1\| \cdots \|A_{L-1}\| \|v\| \left(\sum_{s=1}^{S} \|p_{c,x_s}\|_{\mathcal{H}}^2\right)^{1/2}$$

$$\leq \frac{\|A_1\| \cdots \|A_{L-1}\| \|v\| E(c)}{\sqrt{S}},$$

where the last equality is derived since $p_{c,x}$ is the regularizer that satisfies $\|p_{c,x}\|_{\mathcal{H}} = 1$ for any $x \in \mathcal{X}_0$. $\square$

**Theorem 4.4** *Let $\mathcal{NN}_c = \{NN_c(\mathbf{g}, \cdot) \mid \mathbf{g} \in G^L, |\det W_1|^{-1/2}, \ldots, |\det W_L|^{-1/2} \leq D\}$. The Rademacher complexity bound of $\mathcal{NN}_c$ is*

$$\hat{R}(\mathcal{NN}_c, x_1, \ldots, x_S) \leq \frac{E(c)\|v\| \prod_{l=1}^{L-1} \|A_l\|}{\sqrt{S}} \sup_{|\det W_l|^{-1/2} \leq D} \prod_{l=1}^{L} |\det W_l|^{-1/2}.$$

*Proof.* By Theorem 4.1, we have

$$\hat{R}(\mathcal{NN}_c, x_1, \ldots, x_S) = \frac{1}{S}\mathrm{E}\left[\sup_{\mathbf{g} \in G^L, \, |\det W_l|^{-1/2} \leq D} \sum_{s=1}^{S} NN_c(\mathbf{g}, x_s)\sigma_s\right]$$

$$= \frac{1}{S}\mathrm{E}\left[\sup_{\mathbf{g} \in G^L, \, |\det W_l|^{-1/2} \leq D} \sum_{s=1}^{S} F_c(\mathbf{g}, x_s)|\det W_1|^{-1/2} \cdots |\det W_L|^{-1/2}\sigma_s\right]$$

$$\leq \frac{E(c)\|A_1\| \cdots \|A_{L-1}\| \|v\|}{\sqrt{S}} \sup_{|\det W_l|^{-1/2} \leq D} |\det W_1|^{-1/2} \cdots |\det W_L|^{-1/2}.$$

$\square$

**Theorem 5.1** *Let $\mathcal{F}_c = \{F_c(\theta_1, \ldots, \theta_L, \cdot) \mid |\det W_1^* W_1|^{-1/4}, \ldots, |\det W_L^* W_L|^{-1/4} \leq D\}$. Assume $p_{c,x} \in \mathcal{K}$ for $x \in \mathcal{X}_0$. Let $f_l = v \circ W_L \circ \sigma_{L-1} \circ \cdots \circ W_{l+1} \circ \sigma_l$. Let $\alpha(h) = (\int_{W_l \mathcal{X}_{l-1}} |h(x)|^2 d\mu_{\mathcal{R}(W_l)}(x) / \int_{\tilde{\mathcal{X}}_l} |h(x)|^2 d\mu_{\mathbb{R}^{d_l}}(x))^{1/2}$ for $h \in \tilde{\mathcal{H}}_l$. Then, we have*

$$\hat{R}(\mathcal{F}_c, x_1, \ldots, x_S) \leq \sup_{|\det W_l^* W_l|^{-1/4} \leq D} \frac{E(c)\|v\| \prod_{l=1}^{L-1} \|A_l\| \alpha(f_l)}{\sqrt{S} \prod_{l=1}^{L} |\det W_l^* W_l|^{1/4}},$$

*Proof.* In the same way as Theorem 4.1, we have the same inequality (5) but $\rho(g_l)$ is replaced by $\eta_l(\theta_l) = K_{W_l}$. For $h \in \tilde{\mathcal{H}}_l$, we have

$$\|K_{W_l} h\|^2 = \int_{\mathcal{X}_{l-1}} |h(W_l x)|^2 d\mu_{\mathbb{R}^{d_{l-1}}}(x) = \int_{W_l \mathcal{X}_{l-1}} |h(x)|^2 \frac{1}{|\det W_l^* W_l|^{1/2}} d\mu_{\mathcal{R}(W_l)}(x)$$

$$= \frac{1}{|\det W_l^* W_l|^{1/2}} \frac{\int_{W_l \mathcal{X}_{l-1}} |h(x)|^2 d\mu_{\mathcal{R}(W_l)}(x)}{\int_{\tilde{\mathcal{X}}_l} |h(x)|^2 d\mu_{\mathbb{R}^{d_l}}(x)} \int_{\tilde{\mathcal{X}}_l} |h(x)|^2 d\mu_{\mathbb{R}^{d_l}}(x) = \frac{\alpha(h)^2 \|h\|^2}{|\det W_l^* W_l|^{1/2}} \quad (6)$$

Applying the inequality (6) to the inequality (5) for this case, we obtain the result. □

**Theorem 5.5** *Let $\mathcal{F}_c = \{F_c(\theta_1, \ldots, \theta_L, \cdot) \mid |\det W_1|_{\ker(W_1)^\perp}|^{-1/2}, \ldots, |\det W_L|_{\ker(W_L)^\perp}|^{-1/2} \leq D\}$. Assume $p_{c,x} \in \mathcal{K}$ for $x \in \mathcal{X}_0$. Then, we have*

$$\hat{R}(\mathcal{F}_c, x_1, \ldots, x_S) \leq \sup_{|\det W_l|_{\ker(W_l)^\perp}|^{-1/2} \leq D} \frac{E(c)\|v\| \prod_{l=1}^{L-1} \|A_l\| \alpha(f_l) \prod_{l=1}^{L} \mu_{\ker(W_l)}(\mathcal{Y}_{l-1})}{\sqrt{S} \prod_{l=1}^{L} |\det W_l|_{\ker(W_l)^\perp}|^{1/2}}.$$

*Proof.* For $h \in \tilde{\mathcal{H}}_l$, we have

$$\|\tilde{K}_{\psi_l, W_l} h\|^2 = \int_{\mathcal{X}_{l-1}} |h(W_l x) \psi_l(x)|^2 dx = \int_{\mathcal{Z}_{l-1}} |h(W_l x)|^2 dx \int_{\mathcal{Y}_{l-1}} |\psi_l(x)|^2 dx$$

$$= \int_{W_l \mathcal{X}_{l-1}} |h(x)|^2 \frac{1}{|\det W_l|_{\ker(W_l)^\perp}|} d\mu_{\mathcal{R}(W_l)}(x) \cdot \mu_{\ker(W_l)}(\mathcal{Y}_{l-1})$$

$$\leq \frac{\int_{W_l \mathcal{X}_{l-1}} |h(x)|^2 d\mu_{\mathcal{R}(W_l)}(x)}{|\det W_l|_{\ker(W_l)^\perp}| \int_{\tilde{\mathcal{X}}_l} |h(x)|^2 d\mu_{\mathbb{R}^{d_l}}(x)} \int_{\tilde{\mathcal{X}}_l} |h(x)|^2 d\mu_{\mathbb{R}^{d_l}}(x) \cdot \mu_{\ker(W_l)}(\mathcal{Y}_{l-1})$$

$$= \frac{\|h\|^2 \alpha(h)^2 \mu_{\ker(W_l)}(\mathcal{Y}_{l-1})}{|\det W_l|_{\ker(W_l)^\perp}|}. \quad (7)$$

Applying the inequality (7) to the inequality (5) for this case, we obtain the result. □

**Proposition 5.7** *Let $\mathcal{F}_c = \{F_c(\theta_1, \ldots, \theta_L, \cdot) \mid |\beta(\theta_1)|^{-1/2}, \ldots, |\beta(\theta_L)|^{-1/2} \leq D\}$. Assume $p_{c,x} \in \mathcal{K}$ for $x \in \mathcal{X}_0$. Then, we have*

$$\hat{R}(\mathcal{F}_c, x_1, \ldots, x_S) \leq \sup_{|\beta(\theta_l)|^{-1/2} \leq D} \frac{E(c)\|v\| \prod_{l=1}^{L-1} \|A_l\| \mu_{\ker(P_l)}(\hat{\mathcal{Y}}_l)}{\sqrt{S} \prod_{l=1}^{L} |\beta_l(\theta_l)|^{1/2}}.$$

*Proof.* Since the convolution is a linear operator whose eigenvalues are Fourier components, we have $\|\eta_l(\theta_l)\| \leq |\beta_l(\theta_l)|^{-1/2}$. In the same way as the proof of Theorem 5.5, we have

$$\|\tilde{K}_{\psi_l, P_l}\| \leq \frac{\mu_{\ker(P_l)}(\hat{\mathcal{Y}}_l)}{|\det P_l|_{\ker(P_l)^\perp}|^{1/2}} = \mu_{\ker(P_l)}(\hat{\mathcal{Y}}_l),$$

which proves the result. □

Table 1: Comparison of our bound to existing bounds.

| Authors | Rate | Type |
|---|---|---|
| Neyshabur et al. (2015) | $\frac{2^L \prod_{l=1}^{L} \|W_l\|_{2,2}}{\sqrt{S}}$ | Norm-based |
| Neyshabur et al. (2018) | $\frac{L \max_l d_l \prod_{l=1}^{L} \|W_l\|}{\sqrt{S}} \left( \sum_{l=1}^{L} \frac{\|W_l\|_{2,2}^2}{\|W_l\|^2} \right)^{1/2}$ | |
| Golowich et al. (2018) | $\left( \prod_{l=1}^{L} \|W_l\|_{2,2} \right) \min \left\{ \frac{1}{S^{1/4}}, \sqrt{\frac{L}{S}} \right\}$ | |
| Bartlett et al. (2017) | $\frac{\prod_{l=1}^{L} \|W_l\|}{\sqrt{S}} \left( \sum_{l=1}^{L} \frac{\|W_l^T - A_l^T\|_{2,1}^{2/3}}{\|W_l\|^{2/3}} \right)^{3/2}$ | |
| Wei & Ma (2020) | $\frac{(\sum_{l=1}^{L} \kappa_l^{2/3} \min\{L^{1/2}\|W_l - A_l\|_{2,2}, \|W_l - B_l\|_{1,1}\}^{2/3})^{3/2}}{\sqrt{S}}$ | |
| Ju et al. (2022) | $\frac{\sum_{l=1}^{L} \theta_l \|W_l - A_l\|_{2,2}}{\sqrt{S}}$ | |
| Li et al. (2021) | $\|\mathbf{x}\|| \prod_{l=1}^{L} \|W_l\| - 1| + \gamma_{\mathbf{x}} + \sqrt{\frac{c_{\mathcal{X}}}{S}}$ | |
| Arora et al. (2018) | $\hat{r} + \frac{L \max_i \|f(x_i)\|}{\hat{r}\sqrt{S}} \left( \sum_{l=1}^{L} \frac{1}{\mu_l^2 \mu_{l \to}^2} \right)^{1/2}$ | Compression |
| Suzuki et al. (2020) | $\frac{\hat{r}}{\sqrt{S}} + \sqrt{\frac{L}{S}} \left( \sum_{l=1}^{L} \tilde{r}_l (\tilde{d}_{l-1} + \tilde{d}_l) \right)^{1/2}$ | |
| Hashimoto et al. (2024) | $\frac{\|v\|_{H_L}}{\sqrt{S}} \prod_{l=1}^{L} \frac{G_l \|K_{\sigma_l}\|_{H_l} \|W_l\|^{s_l-1}}{\det(W_l^* W_l)^{1/4}}$ | Koopman-based |
| Ours | $\frac{\|v\|_{\mathcal{L}_L}}{\sqrt{S}} \prod_{l=1}^{L} \frac{G_l \|K_{\sigma_l}\|_{\mathcal{L}_l}}{\det(W_l^* W_l)^{1/4}}$ | |

## B  EXPERIMENTAL DETAILS

All the experiments were executed with Python 3.10 and TensorFlow 2.15.

### B.1  VALIDITY OF BOUNDS

We set $W_1$, $W_2$, and $w_3$ as learnable parameters. We set the loss function as the mean squared error and the optimizer as the SGD with a learning rate 0.001. The learnable parameters are initialized with the orthogonal initialization.

### B.2  COMPARISON TO EXISTING BOUNDS

We constructed a network $f(x) = \sigma_4(W_4\sigma(W_3\sigma(W_2\sigma(W_1 x + b_1) + b_2) + b_3) + b_4)$ with dense layers, where $W_1 \in \mathbb{R}^{1024 \times 784}$, $W_2 \in \mathbb{R}^{2048 \times 1024}$, $W_3 \in \mathbb{R}^{2048 \times 2048}$, $W_4 \in \mathbb{R}^{10 \times 2048}$, $b_1 \in \mathbb{R}^{1024}$, $b_2 \in \mathbb{R}^{2048}$, $b_3 \in \mathbb{R}^{2048}$, $b_4 \in \mathbb{R}^{10}$, $\sigma$ is the elementwise smooth Leaky ReLU (Biswas et al., 2022) with $\alpha = 0.1$ and $\mu = 0.5$, and $\sigma_4$ is the softmax. The learnable parameters $W_1, \ldots, W_4$ are initialized by the orthogonal initialization for $l = 1, 2$ and by samples from the truncated normal distribution for $l = 3, 4$, and we used the Adam optimizer (Kingma & Ba, 2015) for the optimizer with a learning rate of 0.001. We set the loss function as the categorical cross-entropy loss. The result in Figure 4 (b) is the averaged value $\pm$ the standard deviation in 3 independent runs.

### B.3  VALIDITY FOR EXISTING CNN MODELS (LENET)

We constructed a 5-layered LeNet with the hyperbolic tangent activation functions and the averaged pooling layers. We set the optimizer as the Adam optimizer with a learning rate of 0.001. The result in Figure 4 (c) is the averaged value $\pm$ the standard deviation in 3 independent runs.

## C  COMPARISON OF THE KOOPMAN-BASED BOUNDS TO EXISTING BOUNDS

We show the summary of the existing bounds and the proposed bound in Table 1. Here, $\kappa_l$ and $\theta_l$ are determined by the Jacobian and Hessian of the network $f$ with respect to the $j$th layer and $W_l$,

Table 2: Notation table

| | |
|---|---|
| $G$ | Locally compact group for parameters |
| $\Theta_l$ | Set of parameters for the $l$th layer |
| $L$ | Number of layers |
| $d_l$ | Width of the $l$th layer |
| $\mathcal{H}$ | Hilbert space for models |
| $\rho$ | Unitary representation of $G$ on $\mathcal{H}$ |
| $K_\sigma$ | Koopman operator with respect to a function $\sigma$ |
| $W_l$ | Weight matrix for the $l$th layer |
| $\sigma_l$ | Activation function for the $l$th layer |
| $A_l$ | Linear operator corresponding to the activation function for the $l$th layer |
| $f$ | Original deep model |
| $F_c$ | Regularized model with a parameter $c$ |
| $\mathcal{F}_c$ | Function class for models |
| $k$ | Positive definite kernel defined as $k((g_1, \ldots, g_L), (\tilde{g}_1, \ldots, \tilde{g}_L)) = \langle f(g_1, \ldots, g_L), f(\tilde{g}_1, \ldots, \tilde{g}_L) \rangle_{\mathcal{H}}$ |
| $\phi$ | Feature map defined as $\phi(\mathbf{g}) = k(\cdot, \mathbf{g})$ |
| $\tilde{\phi}$ | Feature map representing models defined as $\tilde{\phi}(\mathbf{g}) = f(g_1, \ldots, g_L)$, where $\mathbf{g} = (g_1, \ldots, g_L)$ |
| $\mathcal{K}$ | Hilbert space defined as the closure of $\{\sum_{i=1}^{n} c_i \tilde{\phi}(\mathbf{g}_i) \mid n \in \mathbb{N}, \mathbf{g}_i \in G^L, c_i \in \mathbb{C}\}$ |
| $\iota$ | Isomorphism that maps $\tilde{\phi}(\mathbf{g})$ to $\phi(\mathbf{g})$ |

respectively. In addition, $\tilde{r}_l$ and $\tilde{d}_l$ are the rank and dimension of the $j$th weight matrices for the compressed network and $\| \cdot \|_{p,q}$ is the matrix $(p, q)$-norm. We note that although the form of the existing Koopman-based bound and the proposed bound is similar, our bound is applicable to a wider range of deep models, and the factors $G_l$ and $\|K_{\sigma_l}\|$ are more easily evaluated.

## D    NOTATION TABLE

We provide a notation table 2 that summarizes important notation in the main text.

