# OpenReview forum: "Why High-rank Neural Networks Generalize?: An Algebraic Framework with RKHSs"
_ICLR.cc/2026/Conference — ICLR 2026 Poster_

### Official Review · Reviewer_X6tr · 2025-10-28

**Soundness:** 3
**Presentation:** 2
**Contribution:** 2
**Rating:** 4
**Confidence:** 3

**Summary:**

The paper presents a Rademacher complexity bound for deep NNs derived using Koopman operators, group representations and RKHSs. Distinct from other bounds, the bound includes the determinant of the weight matrices in its denominator, and is therefore tighter than comparable bounds in the case of high-rank weight-matrices.

**Strengths:**

1. The paper builds on previous work on Rademacher complexity bounds via Koopman operators, removing restrictions which limited the applicability of the previously derived bounds.
2. The resulting bounds scale much better than comparable bounds when the weight matrices are high-rank.
3. The experimental results appear to back up the claims make in the paper.

**Weaknesses:**

My main problem with this paper is readability. While I understand that the material is intrinsically difficult, it would nevertheless be helpful if more time was spent systematically introducing the various notations/definitions etc that are used. In particular, I found section 5 to be quite heavy going, and section 5.3 in particular was extremely difficult to parse. Perhaps the authors could include a table (even if it is located in the appendices?) presenting the various notations/definitions (and their inter-connections) in a more systematic way so that the reader isn't left constantly scanning back and forward through paragraphs of dense definitions/notations interleaved with prose to remind themselves what something means when it is eventually used?

More generally it seems to me that this is a somewhat incremental improvement on the previous work in Hashimoto et al, but I am open to argument in this regard as this is open to interpretation.

Specific points and questions:

- Lemma 2.5: it seems to me that this result effectively rules out applying the method to ReLU networks as it blows up in the relevant limit. Can you see any way around this?
- Example 3.3: is the limit in line 205(ish) due to $p_{c,x}(y)$ approaching the Dirac-delta as $c \to \infty$?
- Theorem 4.3: is the supremum here actually bounded? The determinant cannot be zero for the invertible case, but it can be arbitrarily small, so the supremum can presumably be arbitrarily large. Also I assume $A_l$ here is the Koopman operator?
- Theorem 5.1: it is confusing to use $\alpha (f_l)$ here before introducing it (in remark 5.3). Also what role does this factor play? Could it just be replaced by $1$ for clarity in the theorem?
- Section 5.3: the first paragraph here is far to mathematically dense and needs to be expanded for clarity. Also it strikes me that many of these are essentially geometric constructs, so perhaps a diagram could help?

**Questions:**

See weaknesses section.

---

> ### Author Response · Authors · 2025-11-19
>
> ## Readability
> - We added Figure 1 to show the overview of the proposed approach and the differences between the proposed and existing Koopman-based bounds.
>
> - We added definitions and detailed explanations about mathematical notations (e.g. $B(\mathcal{H},\mathcal{H})$, $GL(d)$, isomorphism, Rademacher variable, and Rademacher complexity).
> We also added Figures 2 and 3 and the corresponding explanations in Sections 5.2 and 5.3 to explain how and why we construct the input and output spaces $\mathcal{X}\_l$, $\mathcal{Y}\_l$, $\mathcal{Z}\_l$, and $\tilde{\mathcal{X}}\_l$ of each layer.
>
> - We added a notation table in Appendix D to summarize important notation.
>
> ## Applicability to ReLU networks
> Although applying our result to ReLU networks is future work, we are now seeking the possibility of using the derivative of ReLU.
> Since the derivative of $ReLU(x)=0$ ($x<0$), $ReLU(x)=x$ ($x\ge 0$) is $0$ for $x<0$, the integral $\int_{\mathbb{R}}ReLU'(x) h(ReLU(x)) dx$ is finite for $h\in L^2(\mathbb{R})$, where $ReLU'$ is the derivative of ReLU.
> Thus, by considering the weighted Koopman operator $K_{ReLU',ReLU}$, we may deal with ReLU.
> We will continue investigating this direction and try to find a way to apply our result to ReLU networks.
>
> Please note that even if a network includes ReLU functions, it can be invertible in some cases, and we can apply our results in these cases.
> For example, in the coupling layers in invertible neural networks, we can apply ReLU functions.
> In this case, the transformation is invertible, and we can represent this transformation using a bounded Koopman operator.
>
> ## Limit of $p_{c,x}$
> As you point out, $p_{c,x}$ goes to the Dirac delta function.
> We added the explanation about this in Example 3.3.
>
> ## Supremum in the bound
> Since the function space $\mathcal{F}_c$ is defined as the models with weight matrices whose inverses of determinants are bounded by $D$, the bound is bounded.
> Imposing some boundedness regarding weight matrices is a usual situation to make the supremum in the bound bounded.
>
> ## $\alpha(f_l)$ in Theorem 5.1
> To bound $\alpha(f_l)$, we need the assumption $a\le \vert f_l(x)\vert^2\le b$ for some $a,b>0$.
> Thus, leaving the factor $\alpha(f_l)$, we can obtain a more general statement.
> This factor depends on how large we set $\tilde{\mathcal{X}}\_l$ compared with $W_l\mathcal{X}\_{l-1}$.
> We moved the definition of $\alpha$ before Theorem 5.1 and added the above explanation.
>
> ## About Section 5.3
> We added Figures 2 and 3 and the corresponding explanations in Sections 5.2 and 5.3 to explain how and why we construct the input and output spaces $\mathcal{X}\_l$, $\mathcal{Y}\_l$, $\mathcal{Z}\_l$, and $\tilde{\mathcal{X}}\_l$ of each layer.

---

### Official Review · Reviewer_DC2X · 2025-10-31

**Soundness:** 4
**Presentation:** 3
**Contribution:** 3
**Rating:** 8
**Confidence:** 4

**Summary:**

This paper proposes a new Koopman-operator-based Rademacher complexity bound to explain why neural networks with high-rank weight matrices can generalize well—an observation not captured by existing norm-based or compression-based generalization theories.

The key idea is to construct a reproducing kernel Hilbert space (RKHS) on the parameter space of neural networks using group representations and Koopman operators, enabling a generalization bound applicable to realistic models with bounded input domains and nonsmooth activations (e.g., tanh, sigmoid, Leaky ReLU).

**Strengths:**

Addresses a known theoretical gap:
Most generalization bounds (norm-, compression-, or PAC-Bayes–based) explain why low-rank networks generalize, but fail for empirically observed high-rank networks. This work provides a principled explanation via Koopman-based operator theory.

Mathematical generality:
The approach unifies algebraic, operator-theoretic, and kernel methods, supporting both bounded and nonsmooth activations—something prior Koopman-based bounds (Hashimoto et al., 2024) could not handle.

Elegant use of RKHS:
By defining an RKHS on the parameter space, the authors convert the non-linear deep network mapping into a linear setting where standard Rademacher complexity tools apply — a conceptually clean and general construction.

Practical implications validated by experiments:
The three plots in Figure 1 show consistent improvement in generalization across synthetic regression, MNIST dense nets, and LeNet CNNs when regularization terms derived from the bound are added.

Connections to high-level mathematical structures:
The use of group representations (affine, Heisenberg) and operator algebras (Schur’s Lemma, von Neumann double commutant theorem) to describe neural networks is novel and could inspire deeper theoretical analyses of neural architectures.

**Weaknesses:**

Dependence on bounded activation assumptions:
The proofs assume Koopman operator boundedness (Assumption 2.2), which fails for ReLU (derivative = 0 on half-line).
The authors acknowledge this limitation (Sec. 7) and suggest exploring weighted Koopman operators, but current results exclude the most popular activations.


Interpretability of constants:
The bounds depend on abstract constants that are not easy to estimate in practice. Consequently, the bound is more qualitative than quantitative.

Comparative evaluation:
While it compares against the 2024 Koopman bound, other modern generalization frameworks (e.g., PAC-Bayes, spectral norms, neural tangent kernel analyses) are not benchmarked.

No discussion of sample complexity scaling:
The 1/sqrt(S) dependence mirrors classical Rademacher bounds, but no new scaling exponents are derived.

**Questions:**

1) Can your framework extend to ReLU networks via non-smooth or distributional Koopman operators?

2) How sensitive is the determinant-based bound to weight scaling (e.g., if |det (W)| >> 1 due to scaling, not diversity)?

3) In practical networks, rank deficiency often correlates with overfitting. Does your theory predict the same phenomenon in the inverse direction?

4) How does the bound behave for randomly initialized networks versus trained ones?

5) Could the RKHS kernel you define (Eq. 3.2) be computed empirically for small models to verify the bound numerically?

---

> ### Author Response · Authors · 2025-11-19
>
> Thank you for your constructive comments and positive assessment.
>
> ## Abstract constants in the bound
> Regarding the factor $\alpha(f_l)$, as we mentioned in Remark 5.3, if there exist $a,b>0$ such that $\vert f_l(x)\vert^2$, then it is bounded by 1 by setting $\tilde{\mathcal{X}}\_{l}$ sufficiently large.
> In general case, evaluating this factor is challenging since it depends on the behavior of $f_l$, which is affected by the $l\sim L$th layer.
> As we added in line 348, this value depends on how large we set $\tilde{\mathcal{X}}\_l$ compared with $W_l\mathcal{X}\_{l-1}$.
> Regarding $\mu_{ker(W_{l+1})}(\mathcal{Y}\_l)$, as we mentioned in Remark 5.6, it becomes small if the output of the $l$th layer has small values in the direction of $ker(W_{l+1})$.
> More quantitative evaluation of these factors is future work.
>
> ## Comparison to other modern generalization frameworks
> Please note that our result is a complement of exiting results regarding generalization.
> Since the situations suitable for our bound and existing bounds are different, we cannot simply compare them.
> We will continue investigating the relationship between existing frameworks and our framework.
>
> ## Sample complexity scaling
> Since we applied the fundamental framework of the Radenacher complexity.
> We may extend our framework by using technics such as local Rademacher complexity to derive a rate that is better than $1/\sqrt{S}$, which is future work.
>
> ## Extension to ReLU networks
> Although applying our result to ReLU networks is future work, we are now seeking the possibility of using the derivative of ReLU.
> Since the derivative of $ReLU(x)=0$ ($x<0$), $ReLU(x)=x$ ($x\ge 0$) is $0$ for $x<0$, the integral $\int_{\mathbb{R}}ReLU'(x) h(ReLU(x)) dx$ is finite for $h\in L^2(\mathbb{R})$, where $ReLU'$ is the derivative of ReLU.
> Thus, by considering the weighted Koopman operator $K_{ReLU',ReLU}$, we may deal with ReLU.
> We will continue investigating this direction and try to find a way to apply our result to ReLU networks.
>
> Please note that even if a network includes ReLU functions, it can be invertible in some cases, and we can apply our results in these cases.
> For example, in the coupling layers in invertible neural networks, we can apply ReLU functions.
> In this case, the transformation is invertible, and we can represent this transformation using a bounded Koopman operator.
>
> ## Regarding the weight scaling
> As we discussed in Remark 5.4, there is a tradeoff between $\vert det(W_l)\vert$ and $\sigma_l$ or $\Vert v\Vert$.
> For a scaled weight $cW_l$, the space $\tilde{\mathcal{X}}_l$ for $W_l$ becomes $c\tilde{\mathcal{X}}_l$ for $cW_l$.
> As a result, $\mathcal{X}_l$ also become large and $\sigma_l$ or $\Vert v\Vert$ becomes large.
> Therefore, simply scaling the weight matrices do not always reduce the bound.
>
> ## Relationship with overfitting
> Our bound implies that even for models with high-rank weight matrices, their model complexity can become small.
> Thus, we expect that even for models with high-rank weight matrices, we can avoid the overfitting.
> More formal investigation is future work.
> Since the optimization problem becomes linear problem using the Koopman operator,
> investigating the relationship between the solution of the linear optimization problem and the Koopman-based regularizer may give some anwer to this question.
>
> ##  Behavior of randomly initialized networks
> Randomly initialized weight matrices are full rank with probability 1 [1].
> Also, according to [2], the determinant of a randomly initialized $n$ by $n$ weight matrix tends to be $1/2\log((n-1)!)$.
> Thus, the determinant terms in our bound can be smaller for randomly initialize networks, in particular, for large networks.
> However, as we discussed in Remark 5.4, there is a tradeoff between the determinant terms and the other terms.
> Therefore, the whole bound may not be smaller than the trained networks.
>
> [1] Xinlong Feng and Zhinan Zhang, The rank of a random matrix, Applied Mathematics and Computation, 2007.
>
> [2] Kevin P. Costello, Van Vu, Concentration of random determinants and permanent estimators, SIAM Journal on Discrete Mathematics, 2009.

---

> > ### Author Response · Authors · 2025-11-19
> >
> > ## Empirical computation of the RKHS kernel
> > Please note that $k(\theta_1,\ldots,\theta_L,\theta_1,\ldots,\theta_L)=\Vert f(\theta_1,\ldots,\theta_L)\Vert_{\mathcal{H}\_0}^2=\Vert \tilde{\phi}(\theta)\Vert_{\mathcal{H}\_0}^2$, where $f(\theta_1,\ldots,\theta_L)=\tilde{\phi}(\theta)$ is the model with parameters $\theta_1,\ldots,\theta_L$.
> > Since $\mathcal{H}\_0$ is typically the $L^2$ space, by approximating the integral, we can numerically compute $\Vert \tilde{\phi}(\theta)\Vert_{\mathcal{H}\_0}^2$.
> > The factors in the bound is originally from $\Vert \tilde{\phi}(\theta)\Vert$ (please see the proof of Theorem 4.1).
> > Thus, we can also derive a bound with the form $\Vert \tilde{\phi}(\theta)\Vert E(c)/\sqrt{S}$ (although this bound does not have any interpretation regarding the relationship among the generalization, the weight matrices, and th activation functions).
> > By numerically computing $\Vert \tilde{\phi}(\theta)\Vert_{\mathcal{H}_0}^2$, we can numerically compute the bound.

---

### Official Review · Reviewer_y36o · 2025-11-02

**Soundness:** 1
**Presentation:** 3
**Contribution:** 2
**Rating:** 2
**Confidence:** 3

**Summary:**

The paper develops a new Rademacher complexity bound for deep neural networks by combining Koopman operator theory, group representations, and RKHS methods. Unlike traditional norm- or rank-based bounds, the proposed framework explains why high-rank networks can still generalize well. The authors construct a kernel on the parameter space and prove an isometric correspondence between this Koopman-RKHS and the function space of the network. Overall, the work provides a mathematically elegant perspective on neural network generalization, though some aspects of proofs raise some concerns about their validity.

**Strengths:**

The paper offers a novel and mathematically rigorous framework that unifies Koopman operator theory, RKHS analysis, and group representations for studying neural network generalization. It extends existing Koopman-based approaches to handle nonsmooth activations and bounded domains, broadening their practical relevance.

**Weaknesses:**

The main weakness is a couple of issues that I found in proofs.

**Questions:**

Example 3.3 introduces $p_{c,x}$ which is  Gaussian centered at $x$ whose variance goes to zero, as $c$ goes to infinity.

This function satisfies
$||p_{c,x}||_{1}=1$

but not

$$||p_{c,x}||_{2}=1$$

as claimed in paper.
The latter claim, i.e.
$$||p_{c,x}||_{2}=1$$
is then used in the proof of Theorem 4.3 where the dependence on $c$ vanishes (which I doubt is correct).

---

> ### Author Response · Authors · 2025-11-19
>
> Thank you for your constructive comments.
>
> ## The norm of $p_{c,x}$
> Thank you for pointing out.
> We sincerely apologize that there was an error in Example 3.3.
> As you pointed out, the $L^2$ norm of $p_{c,x}$ is not $1$, but $(c/(2\pi))^{d/4}$.
> We corrected the statement in Example 3.3 and modified the assumption regarding $p_{c,x}$ to $\Vert p_{c,x}\Vert^2\le E(c)$ for some $E(c)>0$ depending on $c$.
> We also modified the statements in Theorems 4.1, 4.4, 5.1, 5.5, and Proposition 5.7.
> We added Remark 4.3 regarding the tradeoff between $F_c$ being close to the
> original model $f$ and the constant $E(c)$ becoming large.
> We emphasize that the main purpose of this paper is to figure out the effect of the weight matrices and activation functions on the generalization property.
> Therefore, these modifications do not affect the main contribution of this paper.
> Thank you again for pointing out the error, which improves the quality of the paper.

---

### Official Review · Reviewer_JANi · 2025-11-05

**Soundness:** 3
**Presentation:** 2
**Contribution:** 3
**Rating:** 4
**Confidence:** 2

**Summary:**

The paper introduces a new bound on the Rademacher complexity of deep neural networks, with the goal of better accounting for the observed good generalization of neural networks with high-rank weight matrices.  The paper improves upon earlier work by Hashimoto et al. 2024 that introduced an approach based on Koopman operators and RKHSs to represent and reason about neural networks.  The main improvement is the analysis of models in a different RKHS than in prior work, thus accounting for a larger and more practical model class, for example including non-smooth activation functions.  A small-scale experiment on MNIST digit classification shows that a regularizer derived from the bound improves on one derived from the earlier bound.

**Strengths:**

- Expanding the reach of existing complexity bounds to a more realistic model class is a very important contribution.

- The style of Koopman-based bounds appears to differ from much of the other existing work studying generalization in neural networks (besides the cited Hashimoto et al. 2024), and so may be complementary to other results.

- The inclusion of empirical evidence, while limited, is helpful.

**Weaknesses:**

- The main weakness is with the presentation.  I fear that in its current form, the paper will be accessible only to an very narrow segment of the ICLR audience.  While I am admittedly not an expert in the recent theoretical ML literature, I believe the paper should be able to make its results and implications much clearer to most of the ICLR community.  By the end of the paper, it is not clear to me in what way exactly the paper improves over Hashimoto et al. 2024, and how this improvement was achieved.  It is possible that parts of my summary above are incorrect as a result (please let me know if so!).  I would have expected a clear statement of the Hashimoto et al. 2024 bound and the new bound, both mathematically and intuitively, and ideally also other existing bounds for comparison.  I also would have expected a more intuitive "walk-through" of the key steps.  There are two paragraphs devoted to introducing RKHSs -- a broadly familiar concept, I believe -- while there are many other more obscure concepts and notation that are relied on but not explained.  While they cannot all be defined for the novice reader, some intuitive explanations of the ideas would go a long way.

- A second possible and more minor weakness (which may be not a weakness at all, and rather due to my own misunderstanding -- again, please let me know if so!) is with the empirical results.  I am not able to understand why the bound was not compared directly to the prior bound (e.g. in Fig. 1(a)) but only as a form of regularization (in Fig. 1(b, c)).  I also wonder why there are no empirical results with more practical, larger datasets and models (say, ImageNet).  It is reasonable for a theoretical paper to limit itself to small experimental settings, if needed.  But it should be stated what are the limitations that prevent scaling up the experiments.

**Questions:**

My remaining questions are mainly about low-level details:

- Regarding these sentences:  "... phenomena in which models with high-rank weight matrices generalize well have been empirically observed (Goldblum et al., 2020).  Since the norm-based and compression-based bounds focus only on the low-rank and nearly low-rank cases, they cannot describe these phenomena."  The second sentence need not be true if the high-rank matrices in Goldblum et al. 2020 are in fact nearly low-rank.  I guess that is not the case, but could you please confirm?

- There is some imprecise language that would be good to clean up.  Examples:
  - "It describes how the model can fit unseen data" -- not clear what "It" refers to.
  - "This bound is described by the ratio of the norm to the determinant of the weight matrix" -- since there are multiple layers and multiple weight matrices, this can't be precise.
  - "the generalization bound is described by the Rademacher complexity" -- unclear what it means for the Rademacher complexity to "describe" a bound.
  - "Our framework fills the gap between the Koopman-based analysis of generalization bounds and practical situations" -- I think this is a bit stronger than intended (i.e. the paper does not fully fill the gap).

- Typo:  "The Hilbert space H to which the modes belong..." --> "... the models belong"

- In Fig. 1(a), I have a hard time seeing the color gradient.  Perhaps a different gradient would be visually clearer.

---

> ### Author Response · Authors · 2025-11-19
>
> Thank you for your constructive comments.
>
> ## Presentation
> - We added the Hashimoto et al. 2024 bound and the proposed bound in Section 1. The formulae are similar, and both the existing and the proposed bounds describe why high-rank neural networks generalize well.
> On the other hand, the difference of the function space $\mathcal{L}\_l$ in the proposed bound from $H_l$ in the existing bound gives a significant benefit to the proposed bound.
> We note that $\mathcal{L}\_l$ is larger than $H_l$, and $\mathcal{L}\_l$ enables us to analyze nonsmooth deep models and bounded data space, which was not possible for the existing bound.
> In addition, it enables us to evaluate the factors $\Vert K_{\sigma_l}\Vert_{\mathcal{L}_l}$ and $G_l$ easily (please see Lemmas~2.3--2.5) and understand the effect of the activation functions on the deep model, which was not possible for the existing bound.
> As a result, the proposed bound significantly improves the existing bound in the sense that it can be applied to a wider range of models and enables us to understand the models well.
> We also added the above explanations in Section 1.
>
> - We added a table that shows existing bounds and the proposed bound in Appendix C.
>
> - We added Figure 1 to show the overview of the proposed approach and the differences between the proposed and existing Koopman-based bounds.
>
> - We added definitions and detailed explanations about mathematical notations (e.g. $B(\mathcal{H},\mathcal{H})$, $GL(d)$, isomorphism, Rademacher variable, and Rademacher complexity).
> We also added Figures 2 and 3 and the corresponding explanations in Sections 5.2 and 5.3 to explain how and why we construct the input and output spaces $\mathcal{X}_l$, $\mathcal{Y}_l$, $\mathcal{Z}_l$, and $\tilde{\mathcal{X}}_l$ of each layer.
>
> ## Empirical results
> - We emphasize that one of the main contribution of this work is that we derive a bound that is applicable to wider range of models than the existing bound, not improving the tightness of the bound.
> In fact, the model in Figure 1(a) is tanh network, and we cannot apply the existing bound to this kind of networks.
> It is possible to compare the behavior as in Figure 1 (a) of the proposed bound with the existing bound using an unrealistic model that is also suitable for the existing bound.
> However, since the main contribution of this work is not improving the tightness of the bound, their behavior will be similar.
>
> - We admit that the empirical results are primitive.
> More complicated models often have additional structures such as batch normalization, dropout, and so on.
> Since this work focus on theory, we applied our bound to the models with fundamental structures that is suitable for our theoretical framework and confirmed whether our theoretical results are valid in these fundamental cases.
> Empirical investigation with more complicated models is future work.
>
> ## Modification of sentences, typos, and Figures
> Thank you for your advice.
> - We corrected the sentence in line 34.
> - We also modified othe sentences according to your advice.
> - We also modified typos and Figure 1 (a).

---

### Author Response · Authors · 2025-11-19
**To all the reviewers**

Thank you bery much for your constructive comments. We carefully addressed your comments and modified the manuscript. The modified parts are colored in red.

We sincerely apologize that there was an error in Example 3.3, as Reviewer y36o pointed out.
We corrected this error and modified the manuscript.
We appreciate Reviewer y36o for pointing out the error, which improves the quality of the paper.
We emphasize that these modifications do not affect the main contribution and main message of this paper.
Please see the answer for Reviewer y36o for more details.

---

### Meta-Review · Area_Chair_cinA · 2026-01-04

**Summary:**

This paper develops a novel Rademacher complexity analysis framework for deep neural networks by combining ideas from Koopman operator theory, group representations, and RKHS, offering a new perspective on why high-rank networks can still generalize well.
The work provides an original and insightful theoretical contribution, though some technical aspects of the proofs and empirical validation remain limited.

All four reviewers—and myself as well—agree that the core idea introduced in this paper is novel, interesting, and elegant.
I thus recommend for acceptance.
But I would be fine if my recommendation were bumped down, as this would give the authors additional time to carefully revise the paper, improve the presentation, further address the remaining concerns, and continue constructive interaction with reviewers.
I believe these could lead to eventual acceptance at a prestigious venue with even more significant contribution.

**Reviewer Concerns:**

- Reviewer **JANi** asked for: (1) better (re)organization and presentation of the results, and (2) further and stronger empirical results. I believe that the first concern has been addressed, at least partially, while the second remains outstanding.
- Reviewer **y36o** pointed out an error in the proof of the main result. This issue has been fixed during the rebuttal, albeit at the cost of introducing an additional term in all bounds.
- Reviewer **DC2X** asked for: (1) extension to ReLU networks, and (2) clarifications of the results and comparisons. I think that some of these points were addressed during the rebuttal, while others were not, but the authors' replies are reasonable.
- Reviewer **X6tr** asked for: (1) improved presentation, (2) extension to ReLU networks, and (3) clarification of the results. I think that some of these were addressed during the rebuttal, while others were not.

**Reviewer Scores:**

- I believe that reviewer **JANi** would have slightly increased or kept the score unchanged (4), as some of the concerns remain outstanding.
- I believe that reviewer **y36o** would have increased the score (2 -> 4 at least, unless the reviewer figured out other issues).
- I believe that reviewer **DC2X** would have kept the score unchanged (8, which is already highly positive), as some concerns were addressed and the authors' responses are generally convincing.
- I believe that reviewer **X6tr** would have slightly increased or kept the score unchanged (4), as some of the concerns remain outstanding.

---

### Decision · Program_Chairs · 2026-01-26

Accept (Poster)